# *AdaSociety*: An Adaptive Environment with Social Structures for Multi-Agent Decision-Making

**Yizhe Huang**[*2,1]    **Xingbo Wang**[*†2]    **Hao Liu**[†3]    **Fanqi Kong**[2,1]    **Aoyang Qin**[4,1]

**Min Tang**[5,1]    **Song-Chun Zhu**[1,2]    **Mingjie Bi**[1]    **Siyuan Qi**[1]    **Xue Feng** [✉1]

[1]State Key Laboratory of General Artificial Intelligence, BIGAI    [2]Peking University
[3]New York University    [4]Tsinghua University    [5]University of Science and Technology of China
szhyz@pku.edu.cn, jacksimbol@stu.pku.edu.cn, fengxue@bigai.ai

## Abstract

Traditional interactive environments limit agents' intelligence growth with fixed tasks. Recently, single-agent environments address this by generating new tasks based on agent actions, enhancing task diversity. We consider the decision-making problem in multi-agent settings, where tasks are further influenced by social connections, affecting rewards and information access. However, existing multi-agent environments lack a combination of adaptive physical surroundings and social connections, hindering the learning of intelligent behaviors. To address this, we introduce *AdaSociety*, a customizable multi-agent environment featuring expanding state and action spaces, alongside explicit and alterable social structures. As agents progress, the environment adaptively generates new tasks with social structures for agents to undertake. In *AdaSociety*, we develop three mini-games showcasing distinct social structures and tasks. Initial results demonstrate that specific social structures can promote both individual and collective benefits, though current reinforcement learning and LLM-based algorithms show limited effectiveness in leveraging social structures to enhance performance. Overall, *AdaSociety* serves as a valuable research platform for exploring intelligence in diverse physical and social settings. The code is available at https://github.com/bigai-ai/AdaSociety.

## 1 Introduction

Classic learning environments [55, 41, 9, 42, 34] have agents trained in small and stationary worlds, which hinders the improvement of agents' intelligence. The learning process stagnates once the environments can no longer provide novel data for agents' explorations. Additionally, agents trained on a fixed task set may suffer from a loss of generalization ability [13]. Single-agent environments [18, 25, 61] set out to solve this problem by constructing adaptive environments that continuously generate new tasks based on agent actions, providing a multitudinous task set.

In multi-agent settings, however, the task set is determined by not only physical surroundings but also social connections among agents. Social connections dramatically impact agents' decision-making by shaping their reward structures and information access [20], and different social structures endow the environments with radically different research problems. For example, centralized scenarios focus on issues like credit assignment and consensus establishment [21, 44], while decentralized settings require agents to address opponent modeling issues and non-stationarity [3, 21, 29, 33].

---

*Equal contribution. † This work was conducted while XW and HL were interns at BIGAI. ✉Corresponding author.

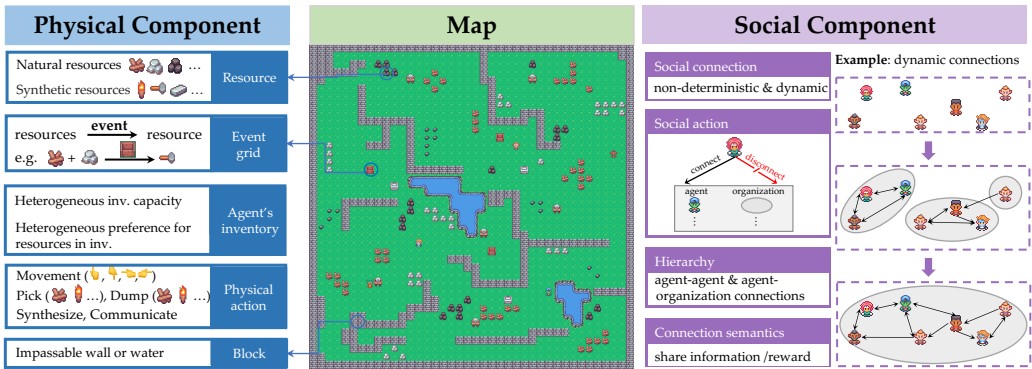

Figure 1: An overview of *AdaSociety*, composed of physical component and social component. **Physical Component** consists of diverse resources and events on the map and heterogeneous agents' inventories. **Social Component** describes the adaptive connections between agents and organizations, which shape information access and reward structure. Agents take social actions to alter their social connections. As shown in the rightmost flowchart, agents are initially independent and can establish individual connections (edges between nodes) and form groups (gray ovals).

What makes the problem even more challenging is that social connections are not predefined but adaptive, which means there's a dynamical interplay between the topology of social connections and agents' states [23]. The adaptive nature of social connections and physical surroundings requires agents to learn continuously, reason about other agents' policies, and balance between physical explorations and establishing social connections. While contemporary multi-agent decision-making environments [6, 2, 53, 66, 48] have achieved great progress in stimulating and testing capabilities of learning algorithms in fixed task sets, they fail to generate new tasks by concurrently considering expanding physical surroundings and adaptive social connections.

To bridge this gap, we propose *AdaSociety*, a multi-agent environment with massive and diverse tasks generated by adaptive social connections and expanding physical surroundings, which are influenced by agents' behavior. In particular, to the best of our knowledge, *AdaSociety* first introduces social states (expressed as a multi-layer directed graph) to explicitly and quantitatively describe the adaptive and dynamic connections between entities, including agents and emerged organizations. This greatly enriches the diversity of tasks, supporting the establishment of stable and long-term relations between entities and the quantitative study of social intelligence, like coalition formation and the emergence of hierarchy. In such an environment, agents need to balance the exploration of physical surroundings and the alteration of social connections, leading to multiple possible victory paths and significant decision-making challenges. To stimulate algorithm design and theoretical analysis in *AdaSociety*, we provide a formulation of the multi-agent decision-making problems, named *Growing-MG* (Sec. 3).

*AdaSociety* serves as a platform for researchers to customize the environment for diverse research needs. Specifically, a set of fundamental elements and mechanisms can be used, and interfaces are provided to set environment attributes and hyper-parameters. Moreover, *AdaSociety* exhibits its characteristics by offering three mini-games, where both tensor- and LLM-based methods are tested.

In summary, this paper makes three contributions. 1) We introduce a novel multi-agent general-sum environment featuring expanding physical surroundings and adaptive social connections. 2) We offer a customizable environment with three built-in mini-games, supporting both tensor- and LLM-based methods. 3) We implement RL and LLM methods in these mini-games and provide preliminary results, laying the groundwork for further research in this environment.

## 2 Environment

### 2.1 Basic Components

The key components of *AdaSociety* (Fig. 1) include the physical component, composed of resources, events, and agents' inventories, and the social component describing connections between agents and organizations. Agents can observe and act to modify both physical and social states.

### 2.1.1 Physical Component

**Resource and Event.** Resources are natural or synthetic. Natural resources scatter randomly on the map. Some natural resources are visible to everyone while others can only be seen when an agent has specific resources in its inventory. For example, only the agent possessing a hammer can observe coal. When agents with specific resources in their inventories stand on an event grid and take the 'synthesize' action, one unit of new resource is synthesized. Synthetic resources will be automatically placed into agents' inventories. These resources and events can be systematically described as a synthesis tree (see Fig. 4). Importantly, agents are unaware of this synthesis tree. They gradually learn the tree through interaction with the environment. Resources, event grids, and agents are initialized in random locations on the map for every episode. While there are existing 3D benchmark environments focusing on perception challenges, our research centers on the domain of multi-agent decision-making. To this end, the map is intentionally crafted in a 2D symbolic format.

**Agent's Inventory.** Every agent has an inventory with maximal capacities of every resource, implying skill diversity. For example, an agent with a $0$ capacity for hammers cannot possess hammers and observe coal. Agents can collect resources from the map into their inventories and dump resources on the map. Agents' rewards are attached to the resources in their inventories, while they exhibit heterogeneity in resource preferences. Specifically, for agent $i$, the reward of resource $\rho$ is $R_i(\rho) = m_i^\rho \cdot h_i(\rho) \cdot \overline{r}^\rho$, where $m_i^\rho$ is the amount of resource $\rho$ in $i$'s inventory, $h_i(\rho) \in \mathbb{R}$ represents $i$'s preference for $\rho$, $\overline{r}^\rho$ is the objective reward of a unit of $\rho$ (see details in Sec. A.1).

### 2.1.2 Social Component

The social component explicitly exhibits the connections between agents or organizations. These connections drastically influence multi-agent decision-making by affecting agents' accessible information and reward structures. Centralization and its complete opposite, decentralization, can be seen as two typical connection structures, presenting very different decision-making problems. *AdaSociety* supports adaptive connections, with corresponding interactions being modeled as general-sum games. *AdaSociety* considers not only the connections between agents but also the subordinate connections between agents and organizations established autonomously by agents. This makes hierarchical connections possible. Agents take social actions to change social states, like connecting or disconnecting with someone. Fig. 1 illustrates evolving connection structures, from fully independent agents to sparsely connected agents with several non-overlapping small groups, and finally to a unified large group. On the other hand, as a customized environment, *AdaSociety* also supports users to predefine and/or fix social connections for their specific research problems. The semantics of connections are diverse, which can be reward sharing, information sharing, or division of labor between involved agents. *AdaSociety* supports that agents negotiate their connection semantics (Sec. 4).

To maintain consistency with the physical component, we refer to these connections between agents and organizations as social states, which are expressed as a multi-layer directed graph (Sec. 3). Social states explicitly and quantitatively express relations between agents or organizations. For example, the cooperation level of two agents can be measured by the frequency of connections between them. Moreover, the combination of social states with successive tasks in *AdaSociety* supports the establishment of stable and long-term relations and the study of social intelligence, like coalition formation and the emergence of hierarchy.

### 2.1.3 Observation and Action

**Observation.** Each agent navigates with a partially observable window, reaching $o$ grids in the four cardinal directions of its current position. Agents can get their own inventory states of collected resources, but not those of co-players. The social states of all the agents are accessible to everyone.

**Action.** Action space consists of social actions and physical actions. Social actions aim to build and break connections with others, including other agents or organizations. Connections are directional. If agent $i$ connects to agent $j$, but not vice versa, $i$ shares its information or reward with $j$, but gets nothing from $j$. Physical actions include movement, picking and dumping specific resources, synthesizing resources on corresponding event grids, and communicating with someone. Newly synthesized resources enrich picking and dumping actions and the action space.

Table 1: Comparison with existing environments. *AdaSociety* is unique for its adaptive connections between entities and expanding game spaces.

| Environment | Multi-agent | Dynamic Spaces | Adaptive Connection | Imperfect Information | Comm. | Multi-task | General Sum | Tensor & LLM |
|---|---|---|---|---|---|---|---|---|
| AI Economist[66] | ✓ | ✗ | ✗ | ✓ | ✗ | ✗ | ✓ | ✗ |
| Boat Race[2] | ✓ | ✗ | ✓ | ✗ | ✗ | ✓ | ✓ | ✗ |
| Crafter[25] | ✗ | ✓ | ✗ | ✓ | ✗ | ✓ | ✗ | ✗ |
| Diplomacy[6] | ✓ | ✗ | ✗ | ✗ | ✓ | ✗ | ✓ | ✓ |
| Melting Pot[2] | ✓ | ✗ | ✗ | ✓ | ✗ | ✓ | ✓ | ✗ |
| MineDojo[18] | ✗ | ✓ | ✗ | ✓ | ✗ | ✓ | ✗ | ✓ |
| Neural MMO[53] | ✓ | ✗ | ✗ | ✓ | ✓ | ✓ | ✗ | ✗ |
| Overcooked[11] | ✓ | ✗ | ✗ | ✗ | ✗ | ✓ | ✗ | ✗ |
| SMAC[48] | ✓ | ✗ | ✗ | ✓ | ✗ | ✗ | ✓ | ✗ |
| Xland[54] | ✓ | ✓ | ✗ | ✓ | ✗ | ✓ | ✓ | ✗ |
| *AdaSociety* | ✓ | ✓ | ✓ | ✓ | ✓ | ✓ | ✓ | ✓ |

## 2.2 Evaluation Metrics

*AdaSociety* provides diverse metrics to evaluate the performances of agents and organizations including **Individual reward**, **Fairness score**, **Completion rate**, and **Average degree** and **Maximum degree** of the social network. Definitions and details of the metrics are discussed in Sec. A.5.

## 2.3 Environment Characteristics

There are various characteristics of *AdaSociety* that make it novel (see Tab. 1). *AdaSociety* is a **multi-agent** decision-making environment, which provides both mini-games for specific research problems and a **customizable** platform to researchers (see details in Sec. A.4). Agents **dynamically connect** with other agents or organizations and autonomously **communicate** to negotiate the semantics of connections, making the emergence of hierarchical social structure and diverse social intelligence possible. With these dynamic and non-deterministic connections, friends may become foes, and vice versa. Thus, the interactions between agents can be modeled as **general-sum games**, where cooperation coexists with competition. Agents navigate this playground with a **partially observable** window centered on their current position. The state and action spaces of *AdaSociety* **dynamically expand**, adapting to agents' (physical and social) behavior. That generates **massive and diverse tasks**, supporting an evaluation of agents' abilities in multiple aspects. *AdaSociety* is **friendly to LLM-** and **tensor-based agents**. We evaluate state-of-the-art RL methods and LLMs in Sec. 5. In addition, we want to stress that the **mutual adaptation between agents and *AdaSociety***, which generates a variety of successive tasks and multiple possible victory paths. Achieving success in *AdaSociety* requires a balance between the exploration of physical components and the alteration of social connections (see Fig. 5). Agents continually learn policies to efficiently explore and achieve goals in *AdaSociety*. Meanwhile, agents' (physical and social) behavior will affect the dynamics of *AdaSociety*. Synthesizing new resources will gradually expand *AdaSociety*'s physical state space and the corresponding physical action space, transition function, and reward function. Updated social states will reshape agents' observation and reward structures. Thus, tasks and task sequences are influenced by agents' behavior and social states, not sampled according to some predefined distribution of tasks. That is to say, *AdaSociety* adapts its tasks and task sequences to agents. Mutual adaptation provides exceptionally massive and diverse complex tasks. The stochasticity and non-stability of *AdaSociety* produce various environment dynamics. Agents need to keep learning to adapt to changing situations.

## 2.4 Research Challenges

As an adaptive multi-agent environment, *AdaSociety* provides a comprehensive platform that presents plenty of research challenges. The adaptive and dynamic characteristics of the physical and social components bring challenges mainly lying in the intricate and unpredictable interactions between agents. Through multi-dimensional **exploration**, agents learn the ability of dynamic environmental **adaptation** and engage in **communication**-enabled interactions. Meanwhile, agents may develop

**social cognition** and utilize this information to conduct **collective reasoning**, which may result in the **emergence** of various behaviors. Details of these challenges are stated in Appendix B.

## 3  Formulation

We now provide a comprehensive definition and analysis of the ***Growing-MG with a social structure***, which are general enough to encompass all the research challenges mentioned above. Three concrete scenarios will be instantiated in next section.

The predominant model in multi-agent sequential decision-making is the Markov Game (MG) [37]. However, a significant limitation of MG is the assumption of constant state and action spaces and unchanged Markovian transitions or rewards, ensuring convergence to some classical solutions such as global optimality or Nash equilibrium [4, 57]. To address dynamic state and action spaces, we introduce two new structures, *Monotonic-MG-bundle* and *Growing-MG* as below. A Growing-MG yields a multi-agent non-stationary decision-making framework. At time step $t$, with state $s_t$ and action $a_t$, the Monotonic-MG-bundle produces $S_{t+1}, A_{t+1}, T_{t+1}, R_{t+1} = \beta(s_t, a_t)$, forming one new MG instance. This framework differs from time-varying games [10, 64, 5], which only model payoff matrix dependent on past actions. On the other hand, both the transition probability and reward function in Growing-MG will evolve triggered with some certain transitions. For simplicity, we denote all possible transition and reward functions on arbitrary state and action space $S, A$, as $\mathcal{T}(S, A) = \{T | T : S \times A \to S\}$ and $\mathcal{R}(S, A) = \{R | R : S \times A \to \mathbb{R}\}$ and the largest possible spaces supported by the environment as universal state space $S_w$ and action space $A_w$.

**Definition 1.** A *base-MG* is a tuple $\mathcal{MG}_b = \langle \mathcal{I}, S_b, A_b, T_b, R_b, \rho, \gamma \rangle$, where $\mathcal{I} = \{1, \ldots, I\}$ is a set of agents; $S_b = \{S_b^1, \ldots, S_b^I\}$ and $A_b = \{A_b^1, \ldots, A_b^I\}$ is the state space and action space of all agents; $T_b : S_b \times A_b \times S_b \mapsto [0, 1]$ and $R_b : S_b \times A_b \mapsto \mathbb{R}^I$ is the transition and reward function; $\rho : S_b \mapsto [0, 1]$ is the initial state distribution and $\gamma$ is the temporal discount factor.

**Definition 2.** *A Monotonic-MG-bundle upon a base-MG $\mathcal{MG}_b$ within the universal state and action space $S_w = \{S_w^1, \ldots, S_w^I\}, A_w = \{A_w^1, \ldots, A_w^I\}$ is a map $\beta : S_t \times A_t \to \{S_{t+1}, A_{t+1}, T_{t+1}, R_{t+1} | S_b^i \subseteq S_t^i \subseteq S_{t+1}^i \subseteq S_w^i, A_b^i \subseteq A_t^i \subseteq A_{t+1}^i \subseteq A_w^i, T_{t+1} \in \mathcal{T}(S_{t+1}, A_{t+1}), R_{t+1} \in \mathcal{R}(S_{t+1}, A_{t+1})\}$.*

**Definition 3.** *A Growing-MG upon a base-MG $\mathcal{MG}_b$ within the universal state and action space $S_w, A_w$ is a tuple $\mathcal{MG}_g = (\mathcal{MG}_b, \beta)$.*

Conceptually, each alteration in the state and action space represents a distinct stage where interrelationships among agents should also change. Inspired by research in complex systems like social sciences and economics [15, 52, 14], we propose enhancing the Growing-MG framework with a multilayer graph structure [26] $\mathcal{G} = (\mathcal{V}, \mathcal{E}, \mathcal{C})$ (see Fig. 6). $\mathcal{C}$ is a set of layers, and $\mathcal{V}$ is the set of nodes in all layers. $\mathcal{E}$ is the set of edges existing between nodes in one layer or neighboring layers. We start with a non-interconnected multiplex system of networks $\{\mathcal{G}^1, \mathcal{G}^2, \cdots, \mathcal{G}^{|\mathcal{C}|}\}$, where each layer $c$ consists of a node set $\mathcal{V}^c$ and an edge set $\mathcal{E}^c$, represented by an adjacency matrix $A_{ij}^c$ with $i, j \in \{1, \cdots, |\mathcal{V}^c|\}$. Nodes in the first layer represent agents in Growing-MG, while higher layers represent groups and hierarchies of groups. To delineate relationship between nodes in neighboring layers such as agent-group membership, we introduce inter-layer connectivity using an adjacency matrix $A_{ij}^{c,c+1}$ with $i \in \{1, \cdots, |\mathcal{V}^c|\}$ and $j \in \{1, \cdots, |\mathcal{V}^{c+1}|\}$. This representation models both static and time-varying networks, as inter-layer and intra-layer connectivity evolves with agents' behavior, distinguishing it from existing multi-agent frameworks that predetermine interactions through reward structures [46, 62]. Finally, we note that both the environmental and social states within the framework can be extended to include observational information [38, 39], thereby further enhancing the framework's generality and practical relevance.

## 4  Mini-games

To provide a comprehensive benchmark and illustrate the characteristics of *AdaSociety*, we propose a set of mini-games (Fig. 2). The three mini-games are arranged in ascending order of the complexity of decision-making. *Social structure*, prescribing agents' partners and connection semantics, evaluates agents' ability to adapt to the changeable social structure. *Contract* predefines connection semantics, where agents need to select partners while learning coordination with various co-players.

In *Negotiation*, agents independently select partners, determine the reward distribution plan with their partners, and behave under the negotiated relationship. All of the three mini-games share the same physical component (Sec. 5.1), which contains a part of the synthesis tree. The following text provides a detailed description of the social components of *Social structure*, *Contract*, *Negotiation*. To show the full complexity of our physical components, another mini-game *Exploration*, which contains all built-in resources and events, is introduced in Sec. C.2.

***Social Structure.*** The explicit representation of *social structure* allows dynamic changes as agents interact with the environment. Pre-defined rules for structure change could be designed to compel agents to alter their social relationships while interacting with the environment. We implement structure change at certain steps: when step $t$ reaches $T_1, T_2, ...,$ the social structures are modified to $\mathcal{G}_1, \mathcal{G}_2, ...,$ respectively. Different categories of social structures are stated in Sec. C.1. This forces agents to learn policies to adapt to the changing social environment.

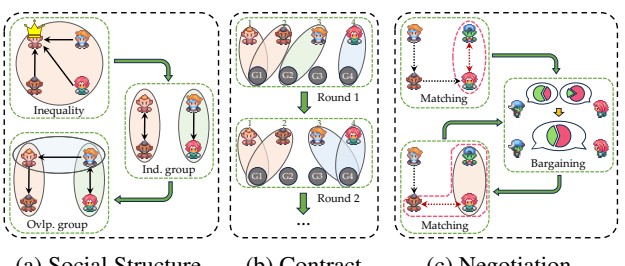

(a) Social Structure     (b) Contract     (c) Negotiation

Figure 2: Overview of three mini-games.

***Contract.*** The environment is divided into two stages: the contract formation stage for determining social connections and the physical interaction stage to interact with the physical component and co-players with determined social connections. The contract formation stage lasts for $cN$ time steps, where $c$ is a positive integer and $N$ is the number of agents, while the physical interaction stage has a duration of $T$. Therefore, the total duration of each episode is $cN + T$. Before the contract formation stage ($0 \leq t < cN$), an order $(i_1, i_2, ..., i_N)$ is randomly sampled. At time $t$, agent $i_k$, where $k = t$ mod $N$, takes social action, selecting a group node $v_g \in V_g$ to connect. An agent can connect with only one group node. Agents within the same group are considered to have formed a contract to share rewards. In the physical interaction stage ($t \geq cN$), all agents act synchronously within the physical component, and the rewards received are equally divided among the agents within the same group.

***Negotiation.*** The game has a negotiation stage followed by a physical stage. In the beginning, agents seek cooperation by selecting an opponent and sending him a request. After mutual requests, agents bargain by exchanging proposals until agreement or breakup. In the bargaining session, agents $i$ and $j$ take turns to perform one of the three actions: (i) PROPOSE a new scheme $(w_i, w_j)$ s.t. $w_i + w_j = 1$, where $w_i$ and $w_j$ represent the partition of rewards obtained by $i$ and $j$ respectively in the physical stage. (ii) ACCEPT the proposal from one's opponent and form a new group (coalition). (iii) DECLINE the proposal and end this session without any commitment. Once a new group is formed, the cooperative relationship between $i$ and $j$ represented by edge $\mathcal{E}_{ij}$ with a payoff distribution $(w_i, w_j)$ is established. Later, when $i$ or $j$ seeks to negotiate with others, it represents the group $\{i, j\}$. For example, if $i$ and an out-group agent $k$ reach a new distribution plan $(w_i^{\text{new}}, w_k^{\text{new}})$, then $k$ is regarded as joining $\{i, j\}$ to form a new group $\{i, j, k\}$ with an updated distribution $(w_i \cdot w_i^{\text{new}}, w_j \cdot w_j^{\text{new}}, w_k^{\text{new}})$.

# 5 Experiments

## 5.1 Environment Setup

We have designed two physical task settings, featuring different levels of difficulty, for *Social Structure*, *Contract*, and *Negotiation*. The parameters of these tasks are provided in Sec. C.3.

In the Easy task, the environment involves a single event HammerCraft. Within this task, agents are categorized into two types based on their inventory capacity and value preference: carpenters and miners. Carpenters have the ability to gather wood and stone, which they can then use to produce hammers through the HammerCraft event. However, their inventory is limited to holding only one hammer at a time. On the other hand, miners are unable to collect stone, making them incapable of producing hammers. However, miners possess the advantage of being able to store a considerable number of hammers in their inventory. Additionally, hammers held by miners are assigned a higher value compared to those held by carpenters.

In the Hard task, the environment becomes more complex with the inclusion of six resources: wood, stone, hammer, coal, torch, and iron, as well as two events: HammerCraft and TorchCraft. Similar to the Easy task, agents are divided into carpenters and miners. Due to the limited capacity of certain resources, only carpenters can execute HammerCraft to produce hammers, while only miners can execute TorchCraft to produce torches. However, carpenters' inventories cannot store coal, which requires a hammer to pick up, and miners' inventories cannot store iron, which requires a torch to pick up. Consequently, in order to maximize group rewards, carpenters and miners should engage in resource exchange, providing the resources they can produce to each other. This collaborative effort ensures that the group can obtain more resources collectively.

## 5.2 Baseline Methods

We use several deep reinforcement learning algorithms as baselines. *Proximal Policy Optimization (PPO)* [49] strikes a balance between sample efficiency and policy stability by constraining policy updates using a trust region approach and a clipped surrogate objective. *RecurrentPPO(RecPPO)* uses PPO for training and add LSTM [28] to maintain memories in the network. *Rainbow* [27] is a value-based method that incorporates several key enhancements into the Deep Q-learning framework. *MAPPO* is the multi-agent version of PPO. It learns a critic that takes the global state and other agents' actions as inputs during training. We employ a convolutional neural network for encoding grid information and a graph convolutional network [32] for encoding social state in all RL methods. The open-source library RLLib [36] is used for RL training.

Additionally, we design a *curriculum learning (CL)* algorithm. It starts with shared rewards to enhance cooperation strategies, then gradually increases social state randomness for learning under different social structures, and finally allows agents to perform social actions to establish their own social state. RecPPO is used for RL training at each stage. We also present a *Large Language Model + rule-based controller (LLM-C)* framework based on GPT-4 [1], which converts environmental information into prompts to query an LLM for high-level plans and then calls a rule-based controller to execute actions based on the generated plan. LLM has been shown to be effective in some single-agent environments, such as MineCraft [59, 56, 58, 67, 60]. The details of the last two algorithms are given in Appendix D.

## 5.3 Results

### 5.3.1 *Social Structure*

In the *Social Structure* mini-game, various static and dynamic social structures are tested to evaluate baseline algorithms. Detailed results are presented in Appendix E. Here, we discuss the result of one **Dynamic** scenario, where the social structure starts with **Inequality**, then switches to **Independent (Ind.) group** at step 30, and alters to **Overlapping (Ovlp.) group** at step 60.

Fig. 3a presents the reward accumulation as agents take actions with three static-structure scenarios and one dynamic-structure scenario, respectively. The results verify the influence of dynamic change in social structure on agent performance since the **Dynamic** curve resembles the **Inequality** scenario initially but then it drops in later steps and approaches **Ovlp. group** scenario.

Fig. 3b and Fig. 3c illustrate the performance of various learning methods. Some traditional methods, such as PPO, RecPPO, and MAPPO, exhibit similar performance, with MAPPO performing worse due

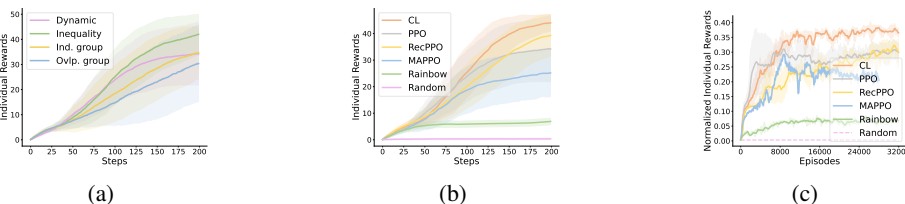

(a)        (b)        (c)

Figure 3: **Dynamic** structure: (a) Individual reward per step with different social structures using 100 samples from PPO-trained policies, (b) Individual reward per step using 100 samples from different policies (c) Learning curves using different learning methods.

Table 2: Average individual reward in *Contract* and *Negotiation*, normalized by the Oracle reward.

| | | CL | PPO | RecPPO | MAPPO | Rainbow | Random |
|---|---|---|---|---|---|---|---|
| *Con.* | Easy | $0.9136_{\pm 0.0023}$ | $0.2286_{\pm 0.0003}$ | $0.2276_{\pm 0.0015}$ | $0.2271_{\pm 0.0003}$ | $0.1987_{\pm 0.0127}$ | $0.0046_{\pm 0.0002}$ |
| | Hard | $0.2773_{\pm 0.0466}$ | $0.1151_{\pm 0.0002}$ | $0.1149_{\pm 0.0000}$ | $0.1137_{\pm 0.0005}$ | $0.0868_{\pm 0.0033}$ | $0.0021_{\pm 0.0000}$ |
| *Nego.* | Easy | $0.3543_{\pm 0.0229}$ | $0.2276_{\pm 0.0006}$ | $0.2278_{\pm 0.0004}$ | $0.2147_{\pm 0.0001}$ | $0.1969_{\pm 0.0105}$ | $0.0040_{\pm 0.0001}$ |
| | Hard | $0.1945_{\pm 0.0109}$ | $0.1093_{\pm 0.0027}$ | $0.1107_{\pm 0.0019}$ | $0.0946_{\pm 0.0032}$ | $0.0905_{\pm 0.0024}$ | $0.0020_{\pm 0.0001}$ |

to the difficulty in learning an effective central critic for heterogeneous agents. Rainbow performs the worst, likely because of its general ineffectiveness in exploration. Curriculum learning demonstrates superior performance by leveraging prior knowledge of different structures to adapt to dynamic scenarios effectively. Additionally, figures in Fig. 3 reveal significant deviations in most tests, regardless of social structures, learning algorithms, or performance metrics. Compared to scenarios without agent groups (Fig. 10a and Fig. 11a), the results indicate that the current algorithms struggle to learn stable policies for scenarios with agent groups.

#### 5.3.2 *Contract*

As depicted in Tab. 2, *Contract* presents a challenge for popular RL methods, as they are stuck in a local equilibrium of completing limited HammerCraft on both tasks (see Fig. 7b), while CL demonstrates notable performance on the Easy tasks and surpasses general RL methods on the Hard tasks. The first curriculum in CL equips the agent with the ability to learn effective policies in the physical realm, and the second curriculum empowers the agent to make informed judgments about different social structures while considering rational physical policies. Ultimately, this knowledge aids CL in selecting an appropriate contract. However, it appears that CL may forget the strategies acquired during the first curriculum, as the reward at the end of the second stage has dropped significantly compared to the end of the first stage (see Tab. 12 for details). This might hamper the performance of CL on the Hard task.

Sharing rewards has been recognized as an effective method for agent groups to acquire cooperative strategies, thereby supporting the feasibility of CL's approach. Fig. 7c and Fig. 7d also illustrates that. In the case of the Easy task, CL eventually establishes a stable group of three individuals who actively share rewards and form a cooperative alliance. However, it is important to note that the size of the group does not directly correlate with high returns. Rainbow, for instance, frequently forms large groups in both tasks but fails to achieve substantial returns. This outcome primarily stems from inherent limitations in the algorithm's learning capabilities.

#### 5.3.3 *Negotiation*

Traditional RL methods struggle to enable carpenters and miners to learn to cooperate through negotiation, dumping some tools to increase the benefit of teammates with larger capacities on the physical stage as shown in Tab. 2. This challenge arises from the complexity of coupling the negotiation and physical stages. Once negotiation fails, dumping tools in the subsequent physical stage would substantially reduce the agents' rewards. Meanwhile, the complex negotiation process exacerbates the convergence problem in multi-agent settings, and agents have the incentive to claim a larger share for themselves to exploit the co-players in bargaining, posing challenges to reaching a consensus agreement. Consequently, in both Easy and Hard tasks, the average and maximum degrees are low, with most agents opting to complete tasks independently, leading to low completion rates in HammerCraft and even a complete failure in TorchCraft (Fig. 8). In the Easy task, miners' rewards heavily rely on carpenters' cooperation, which severely compromises fairness. In contrast, by first learning the optimal strategies in physical environments under different social structures, CL can identify structures with higher cooperation degrees as more beneficial, facilitating consensus during negotiation learning and achieving higher group rewards, fairness, and successful TorchCraft. Additionally, we show the Carpenters/Miners (abbreviated as C/M) split ratio when the negotiation stage is done, which is computed by $\sum_{i \in \{\text{Carpenters}\}} w_i / \sum_{i \in \{\text{Miners}\}} w_j$. All results exceed 1, aligning with the intuition that miners are disadvantaged in negotiations as they cannot independently produce the more rewarding hammers.

### 5.3.4 LLM-C in *AdaSociety*

LLM-C runs three times for each task. Tab. 3 and Tab. 9 presents the quantitative results across various metrics. Benefiting from the embedded commonsense reasoning and social intelligence of LLMs, LLM-C exhibits outstanding performance in all three mini-games, achieving average rewards nearly surpassing all RL-based methods. After being informed of the game rules and the capability differences between carpenters and miners, LLM-C can accurately recognize the importance of cooperation and swiftly form alliances with other players through negotiation or contract.

During the physical stage, manually coded controllers complement LLM's deficiencies in path planning and position judgment, precisely and efficiently realizing the high-level planning generated by the LLM based on the current social structure and physical environment. However, due to common issues with LLMs

Table 3: Average reward of LLM-C across mini-games.

|      | *Social Structure* | *Contract* | *Negotiation* |
|------|--------------------|------------|---------------|
| Easy | -                  | $0.8433_{\pm 0.1312}$ | $0.8733_{\pm 0.1116}$ |
| Hard | $0.7894_{\pm 0.0444}$ | $0.6499_{\pm 0.1716}$ | $0.6862_{\pm 0.1027}$ |

such as hallucinations, context length limitations, and randomness in outputs, LLM-C does not achieve Oracle performance, and it underperforms compared to CL in *Contract*-Easy, further validating the effectiveness of our proposed CL approach.

## 6 Related Work

**Environments.** Several craft-based environments like Malmo [31], Crafter [25], Minedojo [18] and Conan [61] create dynamic state and action spaces that expand with the agent's exploration, which, however, mainly focuses on single-agent setting. Environments including MAgent [65], XLand [54], and Miniworld [12] provide a set of different and transferrable tasks that build from basic elements, and they are open for customization. Melting Pot [2] contains a set of over 50 MARL learning substrates with limited customizability. Interactive games including AI Economist [66], Overcooked [11], MPE [42], Neural MMO [53], and SMAC [48] place agents in diverse systems allowing them to compete or cooperate. Other examples, such as Diplomacy [6], focus on communication between agents. None of these environments contain both dynamic social connections and adaptive tasks like *AdaSociety*.

**Unsupervised Environment Design (UED).** In the paradigm of UED [16, 40, 30], the environment learns a policy $\Gamma : \Pi \to \Delta(\Theta^T)$, which is a function from agent policy $\Pi$ to the environment's parameters $\Theta^T$. Such a policy will automatically produce a distribution over solvable environments and further support the continued learning of the agent's policy. *AdaSociety* does not implement UED to produce diverse tasks. Unlike UED, *AdaSociety* has no goals or objectives, like most ecological systems, and produces multiple tasks through adaptive social structures and expanding physical surroundings.

**Structured multi-agent systems.** In multi-agent systems, various connections may be formed between agents, and these connections may form certain structures. [17], [51] and [45] focus on finding communication topology for multi-agent coordination. Some research models the locality of interaction and learns a joint value via coordination graphs [24, 8, 35]. Networked MARL [63, 46, 47, 62, 50] learns localized policies on environments where agent interactions are contingent upon their connections within a static graph. We focus on dynamic agent connections which shape agents' rewards and observations, and these connections are modeled as a multi-layer graph.

## 7 Conclusion

We introduce *AdaSociety*, a multi-agent environment featuring expanding physical surroundings and adaptive social connections. The environment is capable of generating multiple tasks in adaptation to agents' behavior. *AdaSociety* is friendly to tensor-based and LLM-based methods. *AdaSociety* provides interfaces supporting superb customization and also offers a set of mini-games with diverse social connections. We test several RL and LLM-based algorithms in mini-games. Preliminary results indicate that *AdaSociety* maintains a rational complexity level for current decision-making methods.

There are some limitations of *AdaSociety* illumining our future work. Human-machine interaction is crucial for the study of multi-agent systems, which is one of our key research objectives in *AdaSociety*. While the environment is temporarily not equipped with human interfaces, the current architecture does support the subsequent development of human-machine interfaces. In addition, our game spaces can be further expanded by introducing survival pressures (need for food, hostile creatures, and so on). These negative losses will penalize undesirable actions, complement the roles of positive rewards in reinforcing desirable behavior, and guide more diverse behavior.

## Acknowledgments and Disclosure of Funding

This work is supported by the National Science and Technology Major Project (No. 2022ZD0114904). This work is also supported by the project "Wuhan East Lake High-Tech Development Zone, National Comprehensive Experimental Base for Governance of Intelligent Society".

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

# A Environment Elements

In this section, we elaborate the environment elements predefined in *AdaSociety* including resources, events, and their dependency.

## A.1 Resources

There are 15 kinds of resources in *AdaSociety*, which can be divided into *Natural Resources* and *Synthesized Resources* based on whether they can be produced through events. Some of the natural resources can only be discovered and gathered by agents with certain resources (denoted by *Requirements*) in their inventories. The details of resources are listed in Tab. 4.

Table 4: Resources predefined in *AdaSociety*. **Synthesized** indicates whether the resource can be crafted through events. **Requirement** is an attribute of *natural* resources (*Synthesized* = False) indicating that the resource is observable and collectible to agents carrying the required resources. **Objective reward** denotes the objective rewards of resources.

| Resource | Wood | Stone | Hammer | Coal | Torch | Iron | Steel | Shovel | Pickaxe | GemMine | Clay | Pottery | Cutter | Gem | Totem |
|---|---|---|---|---|---|---|---|---|---|---|---|---|---|---|---|
| Synthesized | ✗ | ✗ | ✓ | ✗ | ✓ | ✗ | ✓ | ✓ | ✓ | ✗ | ✗ | ✓ | ✓ | ✓ | ✓ |
| Requirement | None | None | - | Hammer | - | Torch | - | - | - | Pickaxe | Shovel | - | - | - | - |
| Objective reward | 1 | 1 | 5 | 2 | 20 | 3 | 30 | 100 | 150 | 4 | 4 | 40 | 100 | 200 | 1000 |

## A.2 Events

There are 9 built-in events in *AdaSociety* as listed in Tab. 5. Each event takes 2 to 3 kinds of resources as input and outputs 1 kind of product. Events can only be observed and executed by agents whose inventories meet the event requirements.

Table 5: Events predefined in *AdaSociety*. The ingredients of each event are covered in **Input**. Most events take 2 or 3 different kinds of input resources. The products are listed in **Output**. **Requirement** denotes the resources an agent needs to carry in its inventory to observe and execute the event.

| f Event | Input1 | Input2 | Input3 | Output | Requirement1 | Requirement2 |
|---|---|---|---|---|---|---|
| HammerCraft | 1Wood | 1Stone | - | 1Hammer | - | - |
| TorchCraft | 1Wood | 1Coal | - | 1Torch | Coal | - |
| SteelMaking | 1Iron | 1Coal | - | 1Steel | Iron | - |
| Potting | 2Clay | 1Coal | - | 1Pottery | Clay | - |
| ShovelCraft | 2Steel | 2Wood | - | 1Shovel | Steel | - |
| PickaxeCraft | 3Steel | 2Wood | - | 1Pickaxe | Steel | - |
| CutterCraft | 2Steel | 3Stone | - | 1Cutter | Steel | - |
| GemCutting | 1GemMine | - | - | 1Gem | Cutter | GemMine |
| TotemMaking | 2Gem | 1Pottery | 1Steel | 1Totem | Gem | - |

## A.3 Synthesis Tree

An illustration of the synthetic tree is shown in Fig. 4, which is used by all the mini-games offered by this paper. In Fig. 4, natural and synthetic resources are depicted within a green circle and blue octagon icons respectively. The solid red arrow line attached by a square event icon links low-level resources to high-level products. The eye icons indicate that some resources can help their owner discover new resources or events.

## A.4 Customization

*AdaSociety* is a versatile multi-agent environment platform that supports extensive customization of various elements, features, and hyper-parameters. Researchers can easily create tailored environments for different objectives without needing to delve into the underlying code.

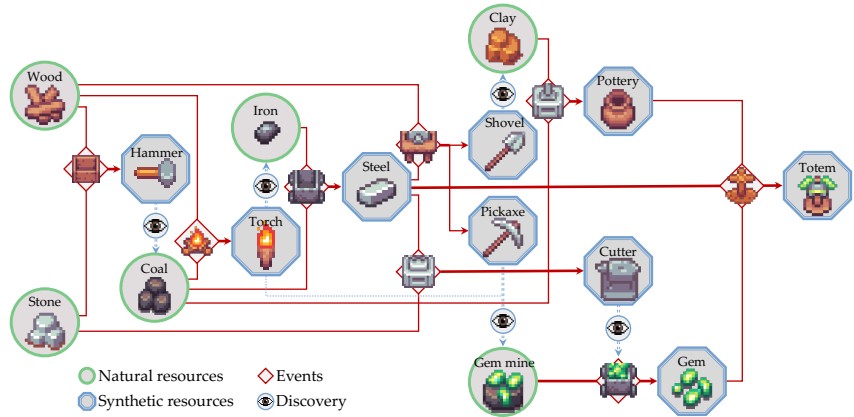

Figure 4: Illustration of a synthesis tree.

Built-in resources and events (See "Synthesis tree" of Fig. 4 and Tab. 5) are included in *AdaSociety* for users to optionally incorporate into their own scenarios. Users are also welcome to define temporary resources and events. The customizable elements and corresponding parameters are listed in Tab. 6.

Table 6: Customizable elements and parameters.

| Element | Parameter | Description |
|---------|-----------|-------------|
| Mapsize | $h, w$ | Map height and map width. |
| Terrain | $B$ | Terrain set $B = \{b_1, \cdots, b_{|B|}\}$. $b_i$ represents a block. |
|  |  | $b_i^{pos}$: the position of block $b_i$ on the map which can be assigned or randomly generated. |
| Resource | $\varrho$ | Set of resources $\varrho = \{\rho_1, \cdots, \rho_{|\varrho|}\}$. Each resource $\rho_i$ has an attribute $\rho_i^{req}$. |
|  |  | $\rho_i^{req}$: Necessary resources in agents' inventories to observe & collect $\rho_i$. |
|  | $\rho_{temp}$ | Temporary resources (Defined by specifying $\rho_{temp}^{req}$) |
| Event | $\mathcal{E}$ | Set of events $\mathcal{E} = \{\epsilon_1, \cdots, \epsilon_{|\mathcal{E}|}\}$. Each event $\epsilon_i$ has attributes $\epsilon_i^{in}, \epsilon_i^{out}, \epsilon_i^{req}$. |
|  |  | $\epsilon_i^{in}$: Resources consumed by event $\epsilon_i$. |
|  |  | $\epsilon_i^{out}$: Resources produced by event $\epsilon_i$. |
|  |  | $\epsilon_i^{req}$: Necessary resources in agents' inventories to observe & execute $\epsilon_i$. |
|  | $\mathcal{E}^{pos}$ | Event positions $\mathcal{E}^{pos} = \{\epsilon_1^{pos}, \cdots, \epsilon_{|\mathcal{E}|^{pos}}\}$. Each $\epsilon_i^{pos}$ represents a list of positions of $\epsilon_i$. |
|  | $\epsilon_{temp}$ | Temporary events (Defined by specifying $\epsilon_{temp}^{in}, \epsilon_{temp}^{out}, \epsilon_{temp}^{req}$) |
| Agent | $P$ | Set of agents $P = \{1, \cdots, |P|\}$ |
|  | $m_i(0)$ | Initial inventories. $m_i^\rho(0)$ denotes the initial number of resource $\rho$ in inventories. |
|  | $i^{cap}$ | Inventory capacity. $i^{cap}: \varrho \to \mathbb{R}$ denotes maximum quantities of resources $i$ can carry. |
|  | $h_i$ | $h_i: \varrho \to \mathbb{R}$ denotes quantities of credits $i$ gets by acquiring resources. |
|  |  | The actual reward obtained by $i$ is $h_i$ multiply by the objective reward of the resource. |
|  | $i^{pos}(0)$ | Initial positions of agents which can be predefined or generated randomly. |

## A.5 Evaluation Metrics

**Individual reward** is calculated as:

$$R_i^c = \sum_{\rho \in \varrho} R_i(\rho), \tag{1}$$

representing agent $i$'s subjective reward of all types of resources $\varrho$.

**Fairness score** is computed based on Gini index [19] to assesses the group-wise fairness:

$$F = 1 - \frac{\sum_{i=1}^{N} \sum_{j=1}^{N} |R_i^c - R_j^c|}{2N \sum_{i=1}^{N} R_i^c}, \tag{2}$$

where $N$ is the number of agents. Intuitively, the greater the value of $F$ a group gets, the fairer it is.

**Individual reward** is one of the most common metrics for decision-making problems. It measures agents' decision-making abilities in maximizing self-interest. However, relying solely on individual rewards can be risky. In general-sum games, agents focus on maximizing their own rewards may engage in shortsighted and exploitative behaviors that harm their own long-term rewards and the collective benefit. For example, in Prisoner's Dilemma, self-interested agents always fall into the inefficient Nash equilibrium of defection, which minimizes one's own reward and the collective benefit. To tackle this issue, we introduce the **fairness score** calculated using the Gini index, which evaluates fairness within a group. In real societies, fairness is a crucial component of social justice, significantly influencing the stability of social structures and the maintenance of long-term cooperation. This metric serves as a reference for selecting agents and algorithms that balance efficiency and fairness, rather than merely pursuing individual gains.

**Completion rate** pertains to the ratio of successful executions of an event to its maximum potential executions. It is computed separately for each event. The **completion rate** is introduced to measure agents' exploration within the synthesis tree. It is calculated as the ratio of actual executions to the optimal executions of the oracle policy (computation of the oracle policy can be found in Supplementary Material). The higher the dimension of the completion rate, the deeper the exploration. Exploration is crucial in RL. The introduction of **completion rate** will guide decision-making algorithms to avoid local optima, actively explore the environment, and find the optimal policy effectively.

**Average degree** of node type $\Gamma \in \{agent, group\}$ is calculated as:

$$\overline{D}_\Gamma = \frac{1}{|\mathcal{N}_\Gamma|} \sum_{n \in \mathcal{N}_\Gamma} D_n, \tag{3}$$

where $\mathcal{N}_\Gamma$ is the set of $\Gamma$ nodes and $D_n$ is the degree of node $n$. **Maximum degree** reflects the maximum degree of a certain type of node, defined as:

$$D_\Gamma^{max} = \max_{n \in \mathcal{N}_\Gamma} D_n. \tag{4}$$

In asymmetric cases (where not all edges are bidirectional), the maximum degree and the average degree mentioned above are calculated separately for in-degrees and out-degrees. Social structure is the distinctive feature of *AdaSociety*. Degree-based metrics, including **average degree** and **maximum degree**, are proposed to describe and measure the topology of social structure, which significantly influences agents' policies and performances by shaping their information streams and reward functions. Agents' degree distribution is generally correlated with their rewards. For example, an agent with a high degree can obtain more information or participate in more reward distribution, thereby gaining higher returns. Combining degree-based metrics with other metrics, like individual reward and fairness, we can recognize the effective social structure for scenarios, guiding the learning of algorithms.

## A.6 Supplementary Figures for Environment Description and Formulation

### A.6.1 Mutual Adaption Between Agents and *AdaSociety*

Based on complex network theory, we say *AdaSociety* is an adaptive environment. In complex network theory, a network is called an adaptive network, if there is a feedback loop between the attributes or behavior of nodes and the topology of the network [22, 43, 7]. In *AdaSociety*, agents build or break connections with others and impact social structure. Conversely, social structure influences agents' observations and reward structures and further influences their attributes and behavior. Thus, following the definition of adaptive networks, the social structure of *AdaSociety* is adaptive. As a key component of *AdaSociety*, social structure influences the generation of new tasks. For example, independent agents collect all kinds of available resources to synthesize high-level resources. However, the team-up agents will be mostly rewarded by collecting or synthesizing some specific kind of resources, according to the division of labor in the team. Furthermore, agents initially can only observe very limited resources (wood and stone in our mini-games) and events (hammercraft). Through exploration in *AdaSociety*, agents gradually discover new resources and events. The appearance of a new kind of resource depends on agents' behavior. For instance, as shown

by the synthesis tree in Fig. 4, which appears next, shovel or cutter, depends on agents' behavior. To sum up, *AdaSociety* is an adaptive environment.

Fig. 5 describes the mutual adaption between agents and *AdaSociety*. To achieve their goals, agents learn policies to adapt their (physical and social) behavior to the environment. Meanwhile, agents' behavior will affect and even change the environment. Specifically, physical actions will expand the physical state space and the corresponding physical action space, reward, and transition functions by synthesizing new resources. Social actions will alter social connections, and then influence agents' information access and reward structures. In *AdaSociety*, there is a feedback loop between agents and the environment, making their coevolution possible and may shed light on the generation of infinite tasks.

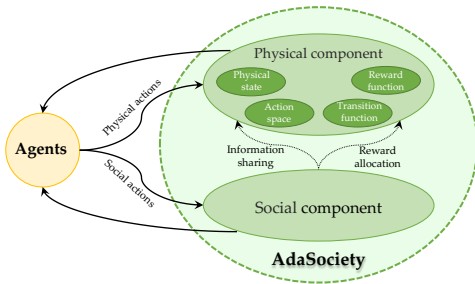

Figure 5: Illustration of the mutual adaptation between agents and *AdaSociety*.

### A.6.2    A Multi-Layer Directed Graph Expression for Social Structure

As shown in Fig. 6, *AdaSociety* expresses social states as a multi-layer directed graph. Each layer shows a level of social organization. *AdaSociety* supports the description of social structures with arbitrary levels, depending on the research problems and the required granularity of social structures. The bottom 0th-level consists of individual agents, who are the fundamental units of decision-making. Nodes in each layer represent entities/agents in the corresponding level. Any agent on the kth-level ($k \geq 1$) is composed of its connected agents on the (k-1)th-level. Its decision-making relies on group norms, like voting, consensus decision-making and delegation. A kth-level agent will affect its (k-1)th-level neighbors' reward functions and observations, thereby influencing their decision-making and enabling their division of labour and cooperation. One agent on the (k-1)th-level may be simultaneously subordinate to any number of agents on the kth-level. For example, an individual employee is the 0th-level agent, a project team composed of several employees is the 1st-level agent, a company consisting of many teams is the 2nd-level agent, and a business group composed of many companies is the 3rd-level agent.

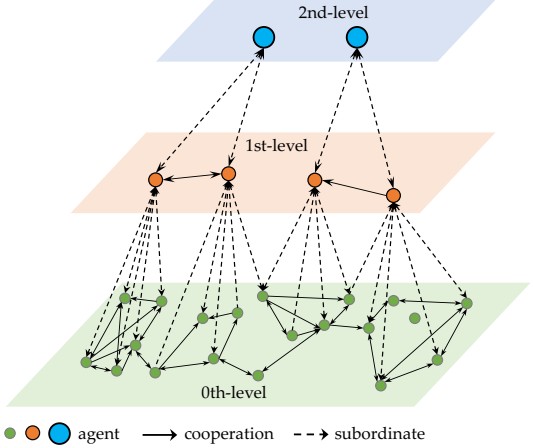

Figure 6: An illustration of social states expressed as a multi-layer directed graph.

*AdaSociety* supports the emergence of high-level social organizations. Edges inside one layer represent cooperative connections, which share information or rewards between involved entities. Edges across layers represent subordinate connections, with low-level entities complying with the policy implemented by the high-level entities. Modeling social states as a multi-layer graph will facilitate the application of existing graph theory knowledge to our research.

## B Research Challenges

**Exploration.** Agents start with a few resources and events within a simple environment initially. As the agents explore the synthesis tree, their behaviors trigger the mechanisms to depict changes in the physical environment. During this process, more resources and events are unlocked gradually, increasing the complexity of the exploration. Dependency between different resources and events evaluates the agents' abilities to make deep explorations in the environment actively.

**Adaptation.** In *AdaSociety*, agents' behaviors could trigger the environment to evolve while the changed environment affects actions that agents can take. Apart from the physical environment, the social structure of the agents could dynamically change as a consequence of either pre-defined rules or agent social behaviors. This requires agents to make decisions accordingly to adapt to and co-evolve with dynamic environments and social relationships.

**Social cognition.** Agents have beliefs in their social structures, which explicitly represent how they interact with other agents, such as exchanging information and sharing rewards. In this complex environment, several achievements require collaboration while resources are limited, forcing agents to cognitively infer others' intentions, evaluate the effectiveness of social structures, and then investigate better choices. This makes *AdaSociety* a suitable environment for studying social cognition and behaviors, such as heterogeneous roles, labor division, ownership, and trust/betrayal.

**Communication.** Portal for communication is provided to agents for sharing information and coordinating actions. A successful agent may learn various communication protocols, context representations, and information processing for optimal objectives. Thus *AdaSociety* could be used for studying the effectiveness of agent communication-enabled interactions, such as negotiation for resource trading, information transitivity, and semantic interoperability.

**Collective reasoning.** Agents are embedded with heterogeneous skills while they only know their own skills. The complex synthesis tree requires agents' abilities to make group decisions on collaboration, such as knowledge sharing and skill transferring for greater group benefit. Additionally, the dynamics of environments make collective reasoning harder, especially for temporal credit assignments. For instance, agents may offer tools (negative immediate reward) to collaborators to exploit unexplored resources (greater delayed reward). Therefore, *AdaSociety* brings challenges for collective reasoning, such as adaptive cooperation and coordination, consensus, and conflict resolution.

**Emergence.** Action space in multiple perspectives, including physical actions, social actions, and communication, enables massive possibilities of agent behaviors without explicit policies. In *AdaSociety*, one could observe the emergence of coordination and cooperation, social structures and norms, and even communication protocols and language.

## C Mini-Game Details

### C.1 Social Structure

As stated in Sec. 3, agents connected by edges share observations, thereby improving their collective situational awareness. Agents connecting to the same group node share rewards based on the edge attributes, incentivizing collaborative efforts to achieve greater rewards. With the social structure, agents act synchronously in the physical environment, following the mechanism for sharing observation and reward defined by the social structure.

In this mini-game, we conducted experiments with static and dynamic social structures. In the static setting, agents are initialized with a certain social structure $\mathcal{G}$ and keep the structure until the end of one episode. In this paper, we categorize social structures that are less than two layers into five types to examine the effects of varying structures on agent behavior and performance: 1) **Isolation:** agents are fully unconnected, i.e., $\mathcal{C} = 1; \mathcal{E} = \emptyset$; 2) **Connection:** agents are connected without forming

groups, i.e., $\mathcal{C} = 1; \mathcal{E} \neq \emptyset$; 3) **Independent Group:** agents are grouped while each agent joins at most one group, i.e., $\mathcal{C} = 2; \sum_j^{\mathcal{V}^2} A_{ij} = 1 \, \forall i \in \mathcal{V}^1$; 4) **Overlapping Group:** agents are grouped and can join multiple groups, i.e., $\mathcal{C} = 2; \exists i \in \mathcal{V}^1, \sum_j^{\mathcal{V}^2} A_{ij} > 1$; 5) **Inequality:** agents are grouped with different reward weights.

In the dynamic setting, pre-defined rules for structure change could be designed to compel agents to alter their social relationships while they take actions within the physical environment. In this task, we design a **Dynamic** scenario, where the social structure starts with **Inequality**, then switches to **Ind. group** at step 30, and alters to **Ovlp. group** at step 60.

### C.2  *Exploration*

In this scenario, all built-in resources and events are included. Physical actions and social actions are available at every step. All agents share a common value preference, where 1) resources near the end of the synthesis tree are assigned high value, and 2) synthetic resources are valued higher than natural resources. Due to partial observation, time limitations, and the challenges associated with exploring new resources, agents may manipulate the social state to encourage interest binding, information sharing, and division of labor, which helps to maximize rewards.

### C.3  Parameters of Mini-Games

The parameters of the Easy task and the Hard task of *Social Structure*, *Contract* and *Negotiation*, along with the parameters of *Exploration* are shown in Tab. 7.

Table 7: Parameters of mini-games. When a resource $\rho$ is not specified, $i^{cap}(\rho)$ defaults to $\infty$ and $h_i(\rho)$ defaults to 1.

| Parameter | Easy | Hard | *Exploration* |
|---|---|---|---|
| $h, w$ | $7, 7$ | $15, 15$ | $20, 20$ |
| $\|B\|$ | 0 | 0 | 25 |
| $b^{pos}$ | - | - | random |
| $\varrho$ | {wood, stone, hammer} | {wood, stone, hammer, coal, torch, iron} | all built-in resources |
| $\mathcal{E}$ | 41 HammerCraft | 98 HammerCraft, 98 TorchCraft | 40 HammerCraft, 40 TorchCraft, 30 SteelMaking, 30 Potting, 20 ShovelCraft, 20 PickaxeCraft, 20 CutterCraft, 10 GemCutting, 10 TotemMaking |
| $\mathcal{E}^{pos}$ | random | random | random |
| $m_i(0)$ | empty | empty | empty |
| $i^{cap}$ | carpenter: {hammer: 1} miner: {wood: 0, stone: 0} | carpenter: {hammer: 1, coal: 0} miner: {stone: 0, torch: 1, iron: 0} | default |
| $h_i$ | carpenter: default miner: {hammer: 2} | carpenter: {coal: 5, torch: 1.5, iron: 20/3} miner: {coal: 5, torch: 1.5, iron: 20/3} | default |
| $i^{pos}(0)$ | random | random | random |

## D  Baseline Details

### D.1  Curriculum Learning

We developed a curriculum learning algorithm for *AdaSociety*. The algorithm controls the social state to promote group cooperation and guides the agent to learn rational physical policies before learning social policies. Our curriculum consists of three stages. We use RecPPO for RL training in each stage.

In the first stage, all individual nodes are compelled to connect to the same group node. This arrangement ensures that all agents belong to the same group. If the reward assigned to an individual increases monotonically with respect to the group reward, which is a common setting, agents in this

stage optimize their actions to enhance the overall benefits. This practical approach encourages the learning of cooperative policies that yield higher rewards, benefiting both individuals and the group.

In the second stage, each individual node is forced to connect to a specific group node with probability $p_K$, while it randomly connects to any of the group nodes with probability $1 - p_K$. The value of $p_K$ gradually decreases with the episode number $K$, resulting in the gradual emergence of diverse social state structures. This setup enables agents to learn physical policies with different social states. During the first two stages, social actions are not allowed, so agents focus solely on learning policies related to physical actions.

Finally, in the third stage, the agent gains the freedom to perform all actions defined in the scenario, thereby acquiring a comprehensive policy for the given task.

### D.2 Large Language Model with Rule-Based Controller

Considering the social nature of *AdaSociety*, we also test the Large Language Model with GPT4 [1] as examples. The LLM agent consists of three modules: observation, reasoning, and execution. In the observation module, we transform the complex physical environment information within the agent's field of view and the current social state into natural language, merging it with the system prompt including game rules and the agent's tasks as inputs. In the reasoning module, the LLM agent generates a high-level plan through few-shot learning, where we require the agent to provide only legal plans that conform to the current environment in the prompt. In the execution module, we decompose the high-level plan into low-level atomic actions through handcrafted functions for interacting with the environment. The process repeats with a new reasoning step to generate a new plan until the current plan is completed. Due to the randomness of LLMs and the uncontrollable nature of multi-agent interactions in *AdaSociety*, it is possible to generate unachievable plans, which will be detected by a monitoring function, prompting the LLM to regenerate the plan.

### D.3 Compute Resources

CPU: 128 Intel(R) Xeon(R) Platinum 8369B CPU @ 2.90GHz; Total memory: 263729336 kB. GPU: 8 NVIDIA GeForce RTX 3090; Memory per GPU: 24576 MiB. Each RL baseline experiment takes 12 to 48 hours, depending on the mini-game and RL algorithm. For LLM-C experiments, each agent takes an average of 5 seconds per step.

## E   Additional Results

This section presents the evaluation results in *Social Structure*, *Contract*, *Negotiation*, and the tests of various baselines in Exploration.

Table 8: Average individual reward in *Exploration*, normalized by the Oracle reward. Traditional RL algorithms can only explore the earlier part of the synthesis tree, resulting in poor returns.

| PPO | RecPPO | MAPPO | Rainbow | Random |
|---|---|---|---|---|
| $0.1744_{\pm 0.0138}$ | $0.1697_{\pm 0.0041}$ | $0.0420_{\pm 0.0123}$ | $0.0051_{\pm 0.0004}$ | $0.0001_{\pm 0.0000}$ |

Table 9: Evaluation results of LLM-C across mini-games.

| | *Social Structure* | *Contract* | | *Negotiation* | |
|---|---|---|---|---|---|
| | Hard | Easy | Hard | Easy | Hard |
| Fairness | $0.8356_{\pm 0.0496}$ | $1.0000_{\pm 0.0000}$ | $1.0000_{\pm 0.0000}$ | $0.7046_{\pm 0.1401}$ | $0.8143_{\pm 0.0350}$ |
| Completion rate (HammerCraft) | $0.6583_{\pm 0.0624}$ | $0.9333_{\pm 0.0236}$ | $1.0000_{\pm 0.0000}$ | $0.8833_{\pm 0.1041}$ | $1.0000_{\pm 0.0000}$ |
| Completion rate (TorchCraft) | $0.6833_{\pm 0.1546}$ | - | $0.9167_{\pm 0.1179}$ | - | $0.9167_{\pm 0.1443}$ |
| Max Group Degree | 4 | $4.000_{\pm 0.000}$ | $8.000_{\pm 0.000}$ | $1.3333_{\pm 0.5773}$ | $3.3333_{\pm 0.5773}$ |
| Avg. Group Degree | 4 | $4.000_{\pm 0.000}$ | $8.000_{\pm 0.000}$ | $1.1667_{\pm 0.2886}$ | $1.625_{\pm 0.6959}$ |

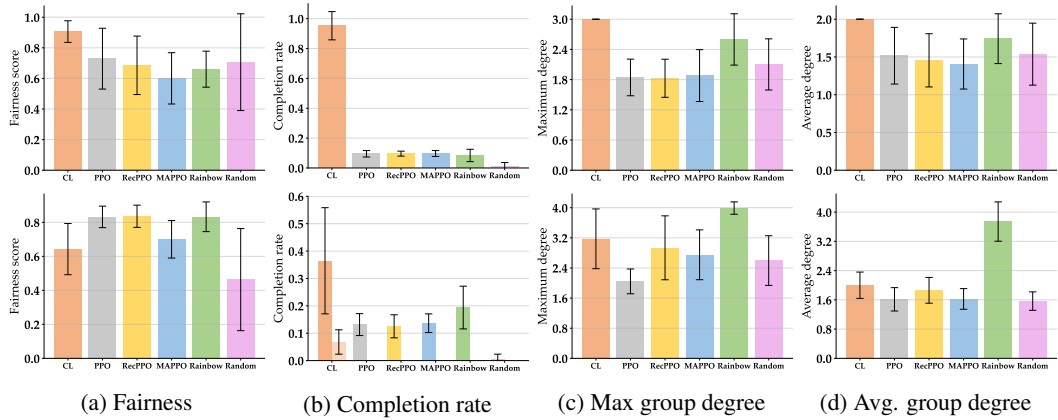

(a) Fairness    (b) Completion rate    (c) Max group degree    (d) Avg. group degree

Figure 7: Evaluation results of *Contract*. Upper row: Easy; lower row: Hard.

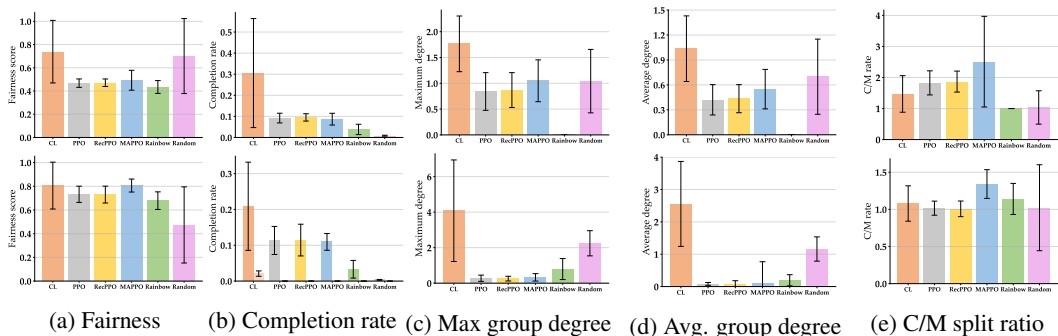

(a) Fairness    (b) Completion rate    (c) Max group degree    (d) Avg. group degree    (e) C/M split ratio

Figure 8: Evaluation results of *Negotiation*. Upper row: Easy; lower row: Hard.

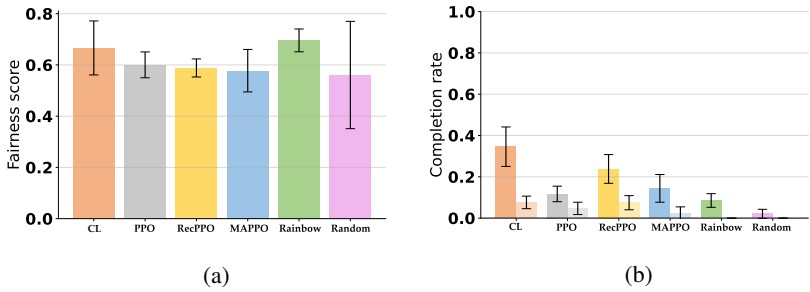

(a)           (b)

Figure 9: *Social Structure*-**Dynamic**: (a) Fairness of agents using different methods, (b) Completion rate of events using different methods.

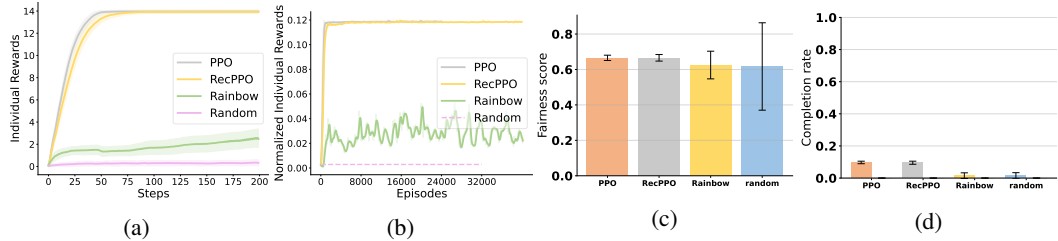

(a)      (b)      (c)      (d)

Figure 10: *Social Structure*-**Isolation**:(a) Individual rewards per step using 100 samples from different policies (b) Learning curves using different methods. (c) Fairness of agents using different methods, (d) Completion rate of events using different methods

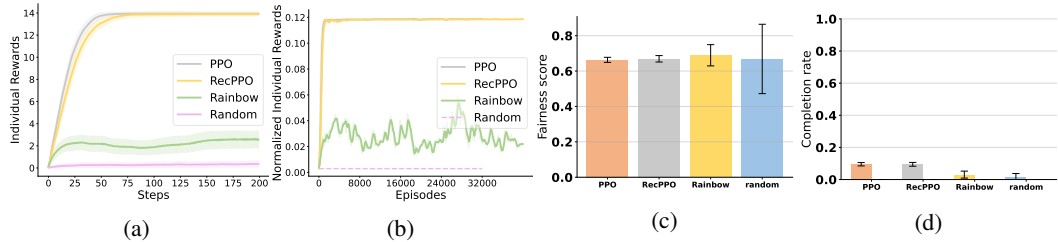

(a)       (b)       (c)       (d)

Figure 11: *Social Structure*-**Connection**: (a) Individual rewards per step using 100 samples from different policies (b) Learning curves using different methods. (c) Fairness of agents using different methods, (d) Completion rate of events using different methods

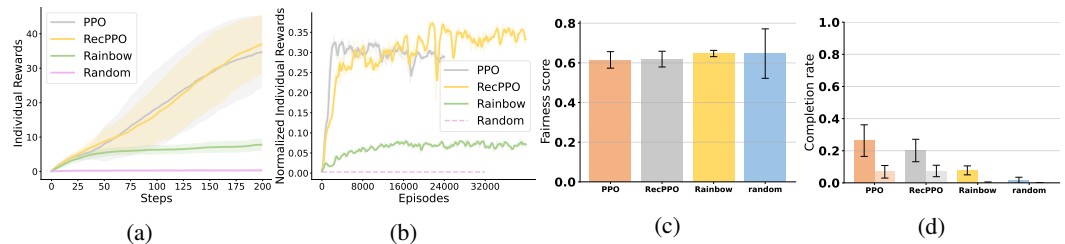

(a)       (b)       (c)       (d)

Figure 12: *Social Structure*-**Independent Group**: (a) Individual rewards per step using 100 samples from different policies (b) Learning curves using different methods. (c) Fairness of agents using different methods, (d) Completion rate of events using different methods

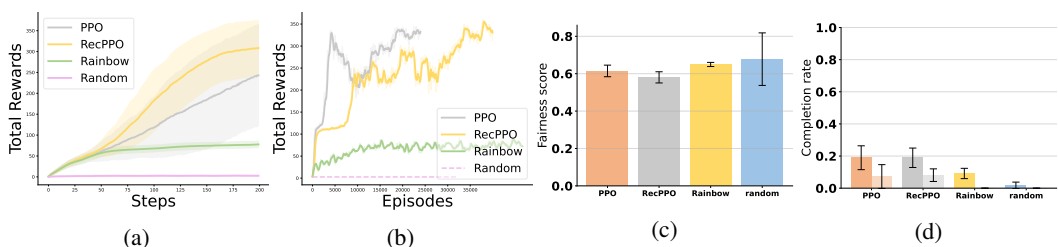

(a)       (b)       (c)       (d)

Figure 13: *Social Structure*-**Overlapping Group**: (a) Individual rewards per step using 100 samples from different policies (b) Learning curves using different methods. (c) Fairness of agents using different methods, (d) Completion rate of events using different methods

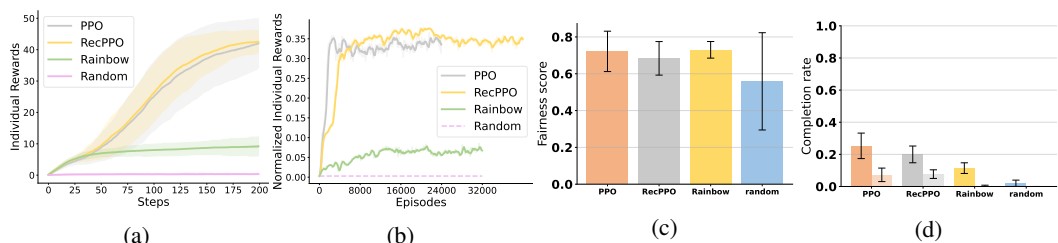

(a)       (b)       (c)       (d)

Figure 14: *Social Structure*-**Inequality**: (a) Individual rewards per step using 100 samples from different policies (b) Learning curves using different methods. (c) Fairness of agents using different methods, (d) Completion rate of events using different methods

Table 10: Time (in hours) and the number of game steps taken by algorithms to converge.

| Time (hours) \| Game Steps | CL | PPO | RecPPO | MAPPO | Rainbow |
|---|---|---|---|---|---|
| Negotiation-Easy | $5.66_{\pm 0.16}$ \| 14M | $0.12_{\pm 0.03}$ \| 0.47M | $1.05_{\pm 0.04}$ \| 2.4M | $21.21_{\pm 0.35}$ \| 1.8M | $14.97_{\pm 0.30}$ \| 42M |
| Negotiation-Hard | $12.96_{\pm 0.39}$ \| 15M | $1.66_{\pm 0.03}$ \| 2.3M | $0.30_{\pm 0.01}$ \| 0.49M | $74.85_{\pm 9.75}$ \| 4.4M | $40.54_{\pm 0.54}$ \| 53M |
| Contract-Easy | $32.03_{\pm 1.38}$ \| 112M | $0.19_{\pm 0.03}$ \| 0.80M | $1.94_{\pm 0.03}$ \| 6.0M | $9.65_{\pm 0.02}$ \| 1.2M | $9.65_{\pm 1.49}$ \| 25M |
| Contract-Hard | $48.58_{\pm 0.56}$ \| 78M | $0.21_{\pm 0.02}$ \| 0.56M | $0.74_{\pm 0.01}$ \| 1.0M | $14.94_{\pm 0.01}$ \| 0.94M | $19.07_{\pm 1.25}$ \| 37M |
| Social Structure - Dynamic | $6.98_{\pm 0.64}$ \| 4.8M | $6.90_{\pm 0.55}$ \| 6.0M | $10.16_{\pm 0.12}$ \| 6.0M | $46.23_{\pm 1.21}$ \| 4.8M | $2.34_{\pm 33.76}$ \| 2.0M |

Table 11: Average number of game steps per second for different player counts (4, 8, 20, 100, 1000). The number of groups is the same as the number of players. The experiment is conducted in *Exploration*, where all built-in resources and events are included (see details in Sec. C.2).

| 4p | 8p | 20p | 100p | 1000p |
|---|---|---|---|---|
| $2495.58_{\pm 26.33}$ | $1245.38_{\pm 3.65}$ | $395.33_{\pm 2.52}$ | $42.58_{\pm 0.26}$ | $2.60_{\pm 0.09}$ |

Table 12: Group rewards in *Contract* after each stage in Curriculum Learning.

| | *Contract-Easy* | *Contract-Hard* |
|---|---|---|
| Stage 1 | $0.9747_{\pm 0.0059}$ | $0.6470_{\pm 0.0313}$ |
| Stage 2 | $0.3435_{\pm 0.0262}$ | $0.2566_{\pm 0.0284}$ |
| Stage 3 | $0.9136_{\pm 0.0023}$ | $0.2773_{\pm 0.0466}$ |

# F  Broader Impact

To contribute to the development of multi-agent decision-making algorithms, we propose *AdaSociety*, a customizable environment with massive and diverse tasks generated by expanding state and action spaces and adaptive social structures. Due to the complexity of tasks and the heterogeneity of agents' capacities and preferences, agents need to team up and even cooperatively establish hierarchical social structures to achieve goals. However, agents may also learn some strategies that are harmful to their co-players, as is common in multi-agent research. We have made significant efforts to mitigate such behaviors through thoughtful design within the environment. Given the heterogeneity among agents and adaptive social structures, harmful behaviors tend to be short-sighted and inferior when it comes to maximizing long-term benefits, with stable cooperation emerging as the optimal strategy. The multiple evaluation metrics introduced in *AdaSociety*, like fairness, also empower researchers to identify and exclude extreme or exploitative agents and facilitate the learning of cooperative behaviors.

Nevertheless, some harmful behaviors may still arise during training. We ask researchers utilizing our platform to meticulously observe agents' behaviors to ensure they align with human values and preferences. Should any misalignment or misrepresentation happen, we encourage contributions to the source code (including but not limited to new evaluation metrics, environmental dynamics or incentive mechanisms) to enhance the platform.

