## Supplementary Material

### Optimal Event Executions for Calculating Completion Rate

When the synthesis tree becomes complicated, it is not straightforward to calculate the maximum potential executions for all the events, making it difficult to evaluate the performance through the metric **Completion rate**. Therefore, we develop an optimization formulation to compute the number of event executions that maximize the credits obtained by agents. This optimization is formulated in a single-agent setting. Since it aims to obtain maximum potential credits, multi-agent cases can also be applied with the set of events being the union of agents' skills. All natural resources can eventually be collected. Tab. 13 shows the parameters and variables used in this optimization.

Table 13: Parameters and variables used in credit optimization.

| Known Parameters | Description |
|---|---|
| $\varrho = R_n \cup R_s$ | Set of resources including natural resources $R_n$ and synthetic resources $R_s$. |
| $m$ | Initial resources. $m_i$ denotes the initial number of natural resources in environments. |
| $h$ | $h_i$ denotes the quantities of credits that an agent gets by acquiring resource $i$. |
| $\mathcal{E}$ | Set of events. Note that $|\mathcal{E}| = |R_s|$ |
| $E$ | Synthesis matrix $E_{|\mathcal{E}|\times|\varrho|}$. Element $E_{ij}$ represents the number of resource $j$ used to synthesize resource $i$ |
| $Q$ | $Q : R_n \to 2^{\mathcal{E}}$ represents the required occurred events to collect certain natural resources |
| $P$ | $P : \mathcal{E} \to \mathbb{R}$ represents the number of produced resources by performing certain events |
| $D$ | $D : \mathcal{E} \to 2^{\mathcal{E}}$ represents the required occurred events to perform certain events. |

| Decision Variables | Description |
|---|---|
| $r$ | Final resources. $r_\rho$ denotes the number of left resource $\rho$ in environments. |
| $\alpha$ | $\alpha_i$ is a binary variable denoting that natural resources $\rho_i$ can be collected. |
| $x$ | Number of occurred events. $x_i$ denotes the number of occurred event $\epsilon_i$ in environments. |
| $\beta$ | $\beta_i$ is a binary variable denoting that event $\epsilon_i$ has occurred in environments. |

$$\max_{r,x,\alpha,\beta} \sum_{i \in R_n} r_i h_i \alpha_i + \sum_{i \in R_s} r_i h_i \beta_i \tag{5a}$$

$$\text{s.t.} \quad r_i = m_i - \sum_{j \in \mathcal{E}} x_j E_{ji}, \forall i \in R_n, \tag{5b}$$

$$r_i = P(i) - \sum_{j \in \mathcal{E}} x_j E_{ji}, \forall i \in R_s, \tag{5c}$$

$$\alpha_i \le x_j, \forall i \in R_n, j \in Q(i) \tag{5d}$$

$$\beta_i \le x_i, \forall i \in \mathcal{E} \tag{5e}$$

$$x_i \le \beta_j \mathcal{M}, \forall i \in \mathcal{E}, j \in D(i) \tag{5f}$$

$$x_i \in \mathbb{N}, \alpha \in \{0, 1\}, \beta \in \{0, 1\} \tag{5g}$$

Eq. 5 presents the optimization formulation, where Eq. 5a calculates the total credits gained by the agents collecting and synthesizing resources; Eq. 5b and Eq. 5c represent the eventually left natural resources and synthetic resources after executing events; Eq. 5d indicates the required events to collect certain resources; Eq. 5e indicates whether a type of event has occurred or not; Eq. 5f states the required events to execute certain events; Eq. 5g limits the values of decision variables.

### Example Prompt for LLM-C

The following examples illustrate the prompts used in LLM-C for each mini-game. The prompts vary slightly for different mini-games and also differ across stages within the same mini-game. Specifically, the prompt for the dynamic scenario in *Social Structure* is presented in Listing 1. For the contract formation stage in *Contract*, the prompt is displayed in Listing 2. Similarly, the prompt for the negotiation stage in *Negotiation* can be found in Listing 3. The physical stage for *Contract* and that for *Negotiation* are the same. There are two physical stage settings, featuring different levels of difficulty. The corresponding prompts are provided in Listing 4 and Listing 5.

Listing 1: Prompt example for dynamic scenario in *Social Structure*.

```
Instructions:
- The AdaSociety game is an open-ended multi-agent environment. The game consists of
     a complex crafting tree, where the agent needs to obtain as many resources as
     possible in the limited time and craft tools to mine more advanced resources to
      maximize its benefit. At the same time, agents can also take other actions to
     help them increase their returns. The numbers of resources are limited.
- Map: AdaSociety is a 2D grid-world game. The map size is 15*15.
    - Natural resources: [Wood, Stone, Coal, Iron]. Some of them can only be
        discovered with some specific tools, which will be introduced next.
    - Tools: [Hammer, Torch]
    - Craft tree:
        - 1 Wood + 1 Stone = 1 Hammer. With a Hammer, Coal can be gathered;
        - 1 Coal + 1 Wood = 1 Torch. With a Torch, Iron can be discovered;
    - All gathered and tools are stored in the agent's inventory.
    - All crafts must be done on the corresponding event grid on the map. For
        example, your inventory must contain wood and stone to craft a hammer.
    - Default amount of all units in crafts is 1.

- Player:
    - There are two kinds of player in the AdaSociety, Carpenters and Miners.
    - The Carpenter can gather many woods, stones and irons and craft hammer, but
        can only own one hammer. The Carpenter CANNOT gather coal so it CANNOT craft
         torch, but its inventory can hold a lot of torches.
    - The Miner can gather many woods and coals, so it can craft torch, but can only
         own one torch. The Miner CANNOT gather stone so it CANNOT craft hammer, but
         its inventory can hold a lot of hammers.
    - For all players, the value of wood is 1, the value of stone is 1, the value of
        hammer is 5, the value of coal is 10, the value of torch is 30, the value
        of iron is 20.
    - Different players may be placed in the same coalition, and the rewards for
        players in the same coalition are split equally, so given the heterogeneity
        between carpenter and miner, players in the same coalition need to cooperate.

Suppose you are a Carpenter named <carpenter_0>. Your aim is to maximize your reward
    , which can gain from the resource value and the craft value.
You can not craft torchs, but you can craft hammers.
At each round in action phase, you will receive the current state:
Step: ...
Current surrounding social environment: ...
Current surrounding physical environment: ...
Your current inventory: ...

You should choose *ONLY ONE* Plan from the following four options: [GATHER <NUM> <
    WOOD/STONE/IRON/TORCH>, CRAFT 1 HAMMER, EXPLORE MAP, DUMP HAMMER]. Here are
    explanations about them:
- GATHER <NUM> <WOOD/STONE/IRON/TORCH>: You shouldn't try to gather items that aren'
    t in your field of view because you don't know where they are. You should also
    not try to gather item that are not in <WOOD/STONE/IRON/TORCH>. You can only
    choose one type of item in your plan.
- CRAFT 1 HAMMER: This plan can help you use the items in your inventory and follow
    the craft tree to craft the resources or tools you need. You can only use this
    plan if you have the corresponding event grid (i.e. the craft point) in your
    view. You should make sure you have enough material to craft.
- EXPLORE MAP: This plan helps you move randomly to explore the map.
- DUMP HAMMER: The plan is to drop hammers on the ground because some agents have
    hammer's capacity of only 1. This action will decrease the corresponding item
    in the inventory by 1. If the item is not in the inventory, please do not
    choose this plan.

<NUM> should be an integer not greater than 10.
Please strictly follow the format above for the output.
```

The response should obey the following format:
Thoughts: Your analysis about your inventory and the current environment.
Plan: One of the above four plans you will take.

Examples:
###
Step: 20
Current surrounding physical environment:
The resources in your observation are: [5 Wood, 5 Stone]. The distances of them are
    [4,6] steps away.
The event grid in your observation are: [Hammer Event]. The distances of them are
    [3] steps away.
You have nothing in your inventory.

Thoughts: I don't have anything in my inventory. There are 5 woods and 5 stones in
    my observation, the wood is closer to me, so I need to gather some wood first.
Plan: GATHER 5 WOOD.
###
Step: 40
Current surrounding physical environment:
The resources in your observation are: [2 Wood, 4 Stone]. The distances of them are
    [8,10] steps away.
The event grid in your observation are: [Hammer Event]. The distances of them are
    [3] steps away.
Your current inventory:
You have 4 Wood, 6 Stone, 0 Hammer.

Thoughts: I have some woods and stones in my inventory but no hammer. There is a
    hammer event in my observation, which means I can craft the hammer.
Plan: CRAFT 1 HAMMER.
###
Step: 60
Current surrounding physical environment:
The resources in your observation are: [1 Wood, 3 Stone]. The distances of them are
    [5,2] steps away.
The event grid in your observation are: [Hammer Event]. The distances of them are
    [3] steps away.
Your current inventory:
You have 2 Wood, 3 Stone, 1 Hammer.

Thoughts: I have some woods and stones, and one hammer in my inventory. Accounting
    for my inventory can only hold one hammer, and there are two miners in my
    coalition who can hold lots hammers, I should dump my hammer to let my
    teammates pick it up, and craft a new one later.
Plan: DUMP HAMMER.
###
Step: 80
Current surrounding physical environment:
The resources in your observation are: [2 Wood, 1 Torch]. The distances of them are
    [2,4] steps away.
The event grid in your observation are: [Hammer Event]. The distances of them are
    [3] steps away.
Your current inventory:
You have 2 Wood, 3 Stone, 1 Hammer.

Thoughts: I have some woods and stones, and one hammer in my inventory but no torch.
     Torch is most valuable tool for me, and my inventory can hold a lot of torches,
     so I need to gather the torch on the map.
Plan: GATHER 1 TORCH.

###
Step: 90
Current surrounding physical environment:

```
The resources in your observation are: []. The distances of them are [] steps away.
    The numbers of them are [] respectively.
The event grid in your observation are: [Hammer Event]. The distances of them are
    [0] steps away.
The people in your observation are: [miner_0], The distances of them are [1] steps
    away.
Your current inventory:
You have NOTHING in your inventory.

Thoughts: I am carpenter_0, and in the current coalition, there are both carpenters
    and miners. The hammercraft event is right next to me. Since I have no wood and
    no stone, I can also not craft hammer. I don't see any resource in my field of
    view, so I need to explore the map to find one.
Plan: EXPLORE MAP.
```

Listing 2: Prompt example for the contract formation stage in *Contract*.

```
Instructions:
- The AdaSociety game is an open-ended multi-agent environment. The game consists of
    a complex crafting tree, where the agent needs to obtain as many resources as
    possible in the limited time and craft tools to mine more advanced resources to
    maximize its benefit. At the same time, agents can also take other actions to
    help them increase their returns, such as negotiating with others to exchange
    resources they need, or forming groups with others to share information and
    rewards.
- Map: AdaSociety is a 2D grid-world game. The map size is 15*15.
    - Natural resources: [Wood, Stone, Coal, Iron]. Some of them can only be
        discovered with some specific tools, which will be introduced next.
    - Tools: [Hammer, Torch]
    - Craft tree:
        - 1 Wood + 1 Stone = 1 Hammer. With a Hammer, Coal can be gathered;
        - 1 Coal + 1 Wood = 1 Torch. With a Torch, Iron can be discovered;
    - All gathered and tools are stored in the agent's inventory.
    - All crafts must be done on the corresponding event grid on the map. For
        example, a Hammer can be crafted ONLY on <Hammer Event>.
    - Default amount of all units in crafts is 1.
    - for carpenter, the value of wood is 1, the value of stone is 1, the value of
        hammer is 5, the value of coal is 10, the value of torch is 30, the value of
        iron is 20.
    - for miner, the value of wood is 1, the value of stone is 1, the value of
        hammer is 5, the value of coal is 10, the value of torch is 30, the value of
        iron is 20.
- Player:
    - carpenter_0: You can pick many woods, stones and irons. You can not pick coal.
        You can own many torchs. Your own inventory can ONLY own 1 hammer.
    - carpenter_1: You can pick many woods, stones and irons. You can not pick coal.
        You can own many torchs. Your own inventory can ONLY own 1 hammer.
    - carpenter_2: You can pick many woods, stones and irons. You can not pick coal.
        You can own many torchs. Your own inventory can ONLY own 1 hammer.
    - carpenter_3: You can pick many woods, stones and irons. You can not pick coal.
        You can own many torchs. Your own inventory can ONLY own 1 hammer.
    - miner_0: You can pick many woods and coals. You can not pick stone and iron.
        You can own many hammers. Your own inventory can ONLY own 1 torch.
    - miner_1: You can pick many woods and coals. You can not pick stone and iron.
        You can own many hammers. Your own inventory can ONLY own 1 torch.
    - miner_2: You can pick many woods and coals. You can not pick stone and iron.
        You can own many hammers. Your own inventory can ONLY own 1 torch.
    - miner_3: You can pick many woods and coals. You can not pick stone and iron.
        You can own many hammers. Your own inventory can ONLY own 1 torch.

Suppose you are a player named <carpenter_0> in the AdaSociety game. You are now in
    the contract phase. Your aim is to maximize your reward, which can gain from
    the resource value and the craft value.
People in a coalition share the rewards equally.
```

```
At each round in contract phase, you will receive the current state:
Step: ...
Round: ...
Current surrounding social environment: ...
Information: ...

In contract phase, you should respond to me with
Thoughts: (Your analysis to the current state)
Action: (About which coalition you want to join)

The <Action> can ONLY be chosen from the following options:
    1. I want to join in Coalition 0.
    2. I want to join in Coalition 1.
    3. I want to join in Coalition 2.
    4. I want to join in Coalition 3.
    5. I want to join in Coalition 4.
    6. I want to join in Coalition 5.
    7. I want to join in Coalition 6.
    8. I want to join in Coalition 7.
Please strictly follow the format above for the output.

Examples:
###
Step: 8
Round: 2
Current surrounding social environment:
coalition 0:[miner_1, carpenter_0, miner_0, carpenter_1, miner_2, miner_3,
    carpenter_2,carpenter_3].
coalition 1:None.
coalition 2:None.
coalition 3:None.
Coalition 4: None
Coalition 5: None
Coalition 6: None
Coalition 7: None
Information: It's carpenter_0's turn.

Thoughts: In this round, all players are currently in Coalition 0. As a carpenter, I
    pick wood and stone but can not own many hammers, while hammer has higher
    reward. Joining a coalition with miners might be beneficial for me to from
    their ability to own more hammers to maximize rewards. Since miners are already
    in Coalition 0, I choose to join in Coalition 0.
Action: I want to join in Coalition 0.
```

Listing 3: Prompt example for the negotiation stage in *Negotiation*.

```
Instructions:
- The AdaSociety game is an open-ended multi-agent environment. The game consists of
    a complex crafting tree, where the agent needs to obtain as many resources as
    possible in the limited time and craft tools to mine more advanced resources to
    maximize its benefit. At the same time, agents can also take other actions to
    help them increase their returns, such as negotiating with others to exchange
    resources they need, or forming groups with others to share information and
    rewards.
- Map: AdaSociety is a 2D grid-world game. The map size is 15*15.
    - Natural resources: [Wood, Stone, Coal, Iron]. Some of them can only be
        discovered with some specific tools, which will be introduced next.
    - Tools: [Hammer, Torch]
    - Craft tree:
        - 1 Wood + 1 Stone = 1 Hammer. With a Hammer, Coal can be gathered;
        - 1 Coal + 1 Wood = 1 Torch. With a Torch, Iron can be discovered;
    - All gathered and tools are stored in the agent's inventory.
    - All crafts must be done on the corresponding event grid on the map. For
        example, a Hammer can be crafted ONLY on <Hammer Event>.
    - Default amount of all units in crafts is 1.
```

```
            - for carpenter, the value of wood is 1, the value of stone is 1, the value of
                hammer is 5, the value of coal is 10, the value of torch is 30, the value of
                iron is 20.
            - for miner, the value of wood is 1, the value of stone is 1, the value of
                hammer is 5, the value of coal is 10, the value of torch is 30, the value of
                iron is 20.
- Player:
    - carpenter_0: You can pick many woods, stones and irons. You can not pick coal.
        You can own many torchs. Your own inventory can ONLY own 1 hammer.
    - carpenter_1: You can pick many woods, stones and irons. You can not pick coal.
        You can own many torchs. Your own inventory can ONLY own 1 hammer.
    - carpenter_2: You can pick many woods, stones and irons. You can not pick coal.
        You can own many torchs. Your own inventory can ONLY own 1 hammer.
    - carpenter_3: You can pick many woods, stones and irons. You can not pick coal.
        You can own many torchs. Your own inventory can ONLY own 1 hammer.
    - miner_0: You can pick many woods and coals. You can not pick stone and iron.
        You can own many hammers. Your own inventory can ONLY own 1 torch.
    - miner_1: You can pick many woods and coals. You can not pick stone and iron.
        You can own many hammers. Your own inventory can ONLY own 1 torch.
    - miner_2: You can pick many woods and coals. You can not pick stone and iron.
        You can own many hammers. Your own inventory can ONLY own 1 torch.
    - miner_3: You can pick many woods and coals. You can not pick stone and iron.
        You can own many hammers. Your own inventory can ONLY own 1 torch.

Suppose you are a player named <carpenter_0> in the AdaSociety game. You are now in
    the first phase: negotiation phase. Your aim is to maximize your reward, which
    can gain from the resource value and the craft value.
Join the coalition to share profits with other members according to the agreed-upon
    distribution ratio.
At each round in negotiation phase, you will receive the current state:
Step: ...
Current surrounding social environment: Specify within {} that it is in an coalition
    .
NegoState: Indicate within [] that two people are negotiating.
Communication log: ...

In negotiation phase, you should respond to me with
Thoughts: (Your analysis to the current state)
Communication: (About who to negotiate with or how to allocate the rewards)

The <Communication> can ONLY be chosen from the following options:
    1. End. I chose to end this bargain.
    2. Accept <PLAYER_NAME>'s proposal. I will gain <NUM>% reward and <PLAYER_NAME>
        will gain <NUM>% reward.
    3. I will make a new proposal. I will propose that I gain <NUM>% reward and <
        PLAYER_NAME> will gain <NUM>% reward.
    4. I will negotiate with <PLAYER_NAME>. I will propose that I gain <NUM>% reward
        and <PLAYER_NAME> will gain <NUM>% reward.
- <PLAYER_NAME> should be from other player names' set: [carpenter_1, carpenter_2,
    carpenter_3, miner_0, miner_1, miner_2, miner_3]
- <NUM> should be an integer which is multiples of ten and is not greater than 100.
Please strictly follow the format above for the output.
!!!If you are in an coalition with someone, you cannot negotiate with them!!!

Examples:
###
Step: 1
Current surrounding social environment:
[{'carpenter_0'}, {'carpenter_1'}, {'carpenter_2'}, {'carpenter_3'}, {'miner_0'}, {'
    miner_1'}, {'miner_2'}, {'miner_3'}]
NegoState:
None.
Communication log:
None.
```

```
Thoughts: I am carpenter_0. miners can craft torch but I can't. As a carpenter, I
    can pick many woods and stones but can only own 1 hammer. miners have a higher
    value for hammers. I have a higher value for torchs. I should negotiate with
    miner to maximize my reward.
Communication: I will negotiate with miner_0. I will propose that I gain 40% reward
    and miner_0 will gain 60% reward.
###
Step 4:
Current surrounding social environment:
[{'carpenter_0'}, {'carpenter_1'}, {'carpenter_2'}, {'carpenter_3'}, {'miner_0'}, {'
    miner_1'}, {'miner_2'}, {'miner_3'}]
NegoState:
['carpenter_1', 'miner_0'],
Communication log:
None

Thoughts: I am carpenter_0. Both miner_0 and carpenter_1 are currently negotiating
    with each other. I can negotiate with miner except for miner_0.
Communication: I will negotiate with miner_1. I will propose that I gain 40% reward
    and miner_0 will gain 60% reward.
###
Step: 10
Current surrounding social environment:
[{'carpenter_0'}, {'carpenter_1'}, {'carpenter_2'}, {'carpenter_3'}, {'miner_0'}, {'
    miner_1'}, {'miner_2'}, {'miner_3'}]
NegoState:
(carpenter_0,miner_0)
Communication log:
miner_0 want to gain 40% reward and you will gain 60% reward.

Thoughts: I am carpenter_0. I'm in negotiate state with miner_0. I can get 60% of
    the reward, which sounds like a good deal and I can accept it.
Communication: Accept miner_0's proposal. I will gain 60% reward and miner_0 will
    gain 40% reward.
```

Listing 4: Prompt example for the Easy task.

```
Instructions:
- The AdaSociety game is an open-ended multi-agent environment. The game consists of
    a complex crafting tree, where the agent needs to obtain as many resources as
    possible in the limited time and craft tools to mine more advanced resources to
    maximize its benefit. At the same time, agents can also take other actions to
    help them increase their returns. The numbers of resources are limited.
- Map: AdaSociety is a 2D grid-world game. The map size is 7*7.
    - Natural resources: [Wood, Stone].
    - Tools: [Hammer]
    - Craft tree:
        - 1 Wood + 1 Stone = 1 Hammer
    - All gathered resources and tools are stored in the agent's inventory.
    - When there are enough resources in the inventory, you can use the CRAFT <TOOL>
        action to synthesize the corresponding tools. For example, your inventory
        must contain wood and stone to craft a hammer.
    - All crafts must be done on the corresponding event grid on the map. For
        example, a Hammer can be crafted ONLY on <Hammer Event>.
    - Default amount of all units in crafts is 1.
    - for carpenter, the value of wood is 1, the value of stone is 1, the value of
        hammer is 5.
    - for miner, the value of wood is 1, the value of stone is 1, the value of
        hammer is 10.
- Player:
    - carpenter_0: can own many woods and stones but can own ONLY own 1 hammer in
        inventory.
    - carpenter_1: can own many woods and stones but can own ONLY own 1 hammer in
        inventory.
```

```
        - miner_0: can NOT own wood and stone, buy can own many hammers in inventory.
        - miner_1: can NOT own wood and stone, buy can own many hammers in inventory.
```

Suppose you are a player named <carpenter_0> in the AdaSociety game. Your aim is to
    maximize your reward, which can gain from the resource value and the craft
    value.
Join the coalition to share profits with other members according to the agreed-upon
    distribution ratio.
At each round in action phase, you will receive the current state:
Step: ...
Current surrounding social environment: ...
payoff: The proportion of the split, shared within an coalition.
Current surrounding physical environment: ...
Your current inventory: ...

In action phase, You should respond to me with
Thoughts: (Your analysis to the current state)
Plan: (The action you plan to take)

You should choose *ONLY ONE* Plan from the following four options: [GATHER <NUM> <
    RESOURCE>, CRAFT 1 <TOOL>, EXPLORE MAP, DUMP <TOOL>]. Here are explanations
    about them:
- GATHER <NUM> <RESOURCE>: RESOURCE is chosen from the Natural resource list above.
    You shouldn't try to gather resources that aren't in your field of view because
     you don't know where they are. You should also not try to gather resources
    that are not natural resources.
- CRAFT 1 <TOOL>: TOOL is chosen from the Tools list above. This plan can help you
    use the items in your inventory and follow the craft tree to craft the
    resources or tools you need. You can only use this plan if you have the
    corresponding event grid (i.e. the craft point) in your view. You should make
    sure you have enough material to craft.
- EXPLORE MAP: This plan helps you move randomly to explore the map.
- DUMP <TOOL>: TOOL is chosen from the Tools list above. The plan is to drop tools
    on the ground because some agents have a tool capacity of only 1. This action
    will decrease the corresponding item in the inventory by 1. If the item is not
    in the inventory, please do not choose this plan.

<NUM> should be an integer not greater than 10.
Please strictly follow the format above for the output.
!!!Before making your crafting choice, please carefully check your inventory to
    ensure you have the necessary materials for crafting. And ensure that the tools
     in the inventory are fewer than the tool capacity. If there are excess tools,
    they should be discarded before crafting new tools. Random crafting selections
    are not allowed!!!
!!!If your inventory don't have hammers, please not dump hammers!!!
!!!craft hammer must need stone and wood, both stone and wood are indispensable.!!!

Examples:
###
Step: 50
Current surrounding social environment:
[{'carpenter_0', 'carpenter_1', 'miner_0', 'miner_1'}]
Current surrounding physical environment:
The resources in your observation are: [Wood, Stone]. The distances of them are
    [5,4] steps away. The numbers of them are [5,4] respectively.
The event grid in your observation are: [Hammer Event]. The distances of them are
    [0] steps away.
The people in your observation are: [miner_1], The distances of them are [1] steps
    away.
Your current inventory:
You have 3 wood.

Thoughts: I'm carpenter_0, and I currently have 3 woods in my inventory. In my
    observation, there is wood and stone nearby, which I can gather. The
    Hammercraft event is also close by, allowing me to craft a hammer. But I hanve
```

```
          no enough material to craft hammer, so I need to gather resources. Since I have
          3 woods, so I need to gather 3 stones.
Plan: GATHER 3 STONE.

###
Step: 90
Current surrounding social environment:
[{'carpenter_0','miner_1'}, {'carpenter_1', 'miner_0'}]
Current surrounding physical environment:
The resources in your observation are: [Wood, Stone]. The distances of them are
          [5,4] steps away. The numbers of them are [4,5] respectively.
The event grid in your observation are: [Hammer Event]. The distances of them are
          [3] steps away.
The people in your observation are: [miner_0, miner_1], The distances of them are
          [3,1] steps away.
Your current inventory:
You have 4 wood, 6 Stone.

Thoughts: I'm carpenter_0. I have 4 wood, 6 Stone. I am in a coalition with both
          carpenters and miners. The resources available are wood and stone, both of
          which are nearby. The hammercraft event is right next to me, allowing me to
          craft a hammer. I currently have more than 1 wood and more than 1 stone in my
          inventory, which is enough to craft a hammer. I have no hammers in the
          inventory. I choose to craft hammer.
Plan: CRAFT 1 HAMMER.

###
Step: 90
Current surrounding social environment:
[{'carpenter_0','miner_1'}, {'carpenter_1', 'miner_0'}]
Current surrounding physical environment:
The resources in your observation are: [Wood, Stone]. The distances of them are
          [5,4] steps away. The numbers of them are [4,5] respectively.
The event grid in your observation are: [Hammer Event]. The distances of them are
          [3] steps away.
The people in your observation are: [miner_0, miner_1], The distances of them are
          [3,1] steps away.
Your current inventory:
You have 4 wood, 6 Stone, 1 hammer.

Thoughts: I'm carpenter_0. I have 4 wood, 6 Stone, 1 hammer. I am in a coalition
          with both carpenters and miners. The resources available are wood and stone,
          both of which are nearby. The hammercraft event is right next to me, allowing
          me to craft a hammer. I currently have more than 1 wood and more than 1 stone
          in my inventory, which is enough to craft a hammer. Since I have one hammer and
           miner_1 who is also in my coalition is closer to me than miner0 in my
          observation, I should consider discarding my current hammer before crafting a
          new one to maximize the coalition's rewards, which miner_1 can pick the hammer
          and if miner_1 own hammer, it will gain more rewards than I own hammers.
Plan: DUMP HAMMER.
```

Listing 5: Prompt example for the Hard task.

```
Instructions:
- The AdaSociety game is an open-ended multi-agent environment. The game consists of
     a complex crafting tree, where the agent needs to obtain as many resources as
     possible in the limited time and craft tools to mine more advanced resources to
      maximize its benefit. At the same time, agents can also take other actions to
     help them increase their returns. The numbers of resources are limited.
- Map: AdaSociety is a 2D grid-world game. The map size is 15*15.
   - Natural resources: [Wood, Stone, Coal, Iron]. Some of them can only be
        discovered with some specific tools, which will be introduced next.
   - Tools: [Hammer, Torch]
   - Craft tree:
       - 1 Wood + 1 Stone = 1 Hammer. With a Hammer, Coal can be gathered;
```

- 1 Coal + 1 Wood = 1 Torch. With a Torch, Iron can be discovered;
        - All gathered and tools are stored in the agent's inventory.
        - All crafts must be done on the corresponding event grid on the map. For
            example, your inventory must contain wood and stone to craft a hammer.
        - Default amount of all units in crafts is 1.
        - for carpenter, the value of wood is 1, the value of stone is 1, the value of
            hammer is 5, the value of coal is 10, the value of torch is 30, the value of
            iron is 20.
        - for miner, the value of wood is 1, the value of stone is 1, the value of
            hammer is 5, the value of coal is 10, the value of torch is 30, the value of
            iron is 20.- Player:
        - carpenter_0: You can gather many woods, stones and irons. You can not gather
            coal. You can own many torchs. Your own inventory can ONLY own 1 hammer.
        - carpenter_1: You can gather many woods, stones and irons. You can not gather
            coal. You can own many torchs. Your own inventory can ONLY own 1 hammer.
        - carpenter_2: You can gather many woods, stones and irons. You can not gather
            coal. You can own many torchs. Your own inventory can ONLY own 1 hammer.
        - carpenter_3: You can gather many woods, stones and irons. You can not gather
            coal. You can own many torchs. Your own inventory can ONLY own 1 hammer.
        - miner_0: You can gather many woods and coals. You can not gather stone and
            iron. You can own many hammers. Your own inventory can ONLY own 1 torch.
        - miner_1: You can gather many woods and coals. You can not gather stone and
            iron. You can own many hammers. Your own inventory can ONLY own 1 torch.
        - miner_2: You can gather many woods and coals. You can not gather stone and
            iron. You can own many hammers. Your own inventory can ONLY own 1 torch.
        - miner_3: You can gather many woods and coals. You can not gather stone and
            iron. You can own many hammers. Your own inventory can ONLY own 1 torch.

Suppose you are a player named <carpenter_0> in the AdaSociety game. You are now in
    the action phase. Your aim is to maximize your reward, which can gain from the
    resource value and the craft value.
You can not craft torchs, but you can craft hammers.
Join the coalition to share profits with other members according to the agreed-upon
    distribution ratio.
At each round in action phase, you will receive the current state:
Step: ...
Current surrounding social environment: ...
payoff: The proportion of the split, shared within an coalition.
Current surrounding physical environment: ...
Your current inventory: ...

In action phase, You should respond to me with
Thoughts: (Your analysis to the current state)
Plan: (The action you plan to take)

You should choose *ONLY ONE* Plan from the following four options: [GATHER <NUM> <
    WOOD/STONE/IRON/TORCH>, CRAFT 1 HAMMER, EXPLORE MAP, DUMP HAMMER]. Here are
    explanations about them:
- GATHER <NUM> <WOOD/STONE/IRON/TORCH>: You shouldn't try to gather items that aren'
    t in your field of view because you don't know where they are. You should also
    not try to gather item that are not in <WOOD/STONE/IRON/TORCH>. You can only
    choose one type of item in your plan.
- CRAFT 1 HAMMER: This plan can help you use the items in your inventory and follow
    the craft tree to craft the resources or tools you need. You can only use this
    plan if you have the corresponding event grid (i.e. the craft point) in your
    view. You should make sure you have enough material to craft.
- EXPLORE MAP: This plan helps you move randomly to explore the map.
- DUMP HAMMER: The plan is to drop hammers on the ground because some agents have
    hammer's capacity of only 1. This action will decrease the corresponding item
    in the inventory by 1. If the item is not in the inventory, please do not
    choose this plan.

<NUM> should be an integer not greater than 10.
Please strictly follow the format above for the output.

!!!Before making your crafting choice, please carefully check your inventory to
    ensure you have the necessary materials for crafting. And ensure that the tools
    in the inventory are fewer than the tool capacity. If there are excess tools,
    they should be discarded before crafting new tools. Random crafting selections
    are not allowed!!!
!!!If your inventory don't have hammers, please do not dump hammers!!!
!!!craft hammer must need stone and wood, both stone and wood are indispensable.!!!

Examples:
###
Step: 50
Current surrounding social environment:
[{carpenter_0, carpenter_1, carpenter_2, carpenter_3, miner_0, miner_1, miner_2,
    miner_3}].
payoff: 0.1, 0.1, 0.1, 0.1, 0.1, 0.1, 0.2, 0.2
Current surrounding physical environment:
The resources in your observation are: [Wood, Stone]. The distances of them are
    [5,4] steps away. The numbers of them are [5,4] respectively.
The event grid in your observation are: [Hammer Event]. The distances of them are
    [0] steps away.
The people in your observation are: [miner_1], The distances of them are [1] steps
    away.
Your current inventory:
You have 3 wood.

Thoughts: I'm carpenter_0, and I currently have 3 woods in my inventory. In my
    observation, there is wood and stone nearby, which I can gather. The
    Hammercraft event is also close by, allowing me to craft a hammer. But I hanve
    no enough material to craft hammer, so I need to gather resources. Since I have
    3 woods, so I need to gather 3 stones.
Plan: GATHER 3 STONE.

###
Step: 90
Current surrounding social environment:
[{'carpenter_0', 'miner_1'}, {'carpenter_1', 'miner_1','miner_2', 'carpenter_2', '
    miner_3', 'carpenter_3'}]
payoff: 0.6, 0.1, 0.1, 0.2, 0.2, 0.4, 0.2, 0.2
Current surrounding physical environment:
The resources in your observation are: [Wood, Stone]. The distances of them are
    [5,4] steps away. The numbers of them are [5,4] respectively.
The event grid in your observation are: [Hammer Event]. The distances of them are
    [0] steps away.
The people in your observation are: [miner_1], The distances of them are [1] steps
    away.
Your current inventory:
You have 4 wood, 6 Stone, 1 hammer.

Thoughts: I'm carpenter_0. In my coalition, there are mostly stones and a minority
    of wood. I can craft hammer heads first to help the coalition gain greater
    profits. miner_1 is in my coalition and he is closer than other people in order
    to my hammer don't be gathered by other coalition, miner own hammers can bring
    more rewards to the coalition, so I will dump hammer.
Plan: DUMP HAMMER.

###
Step: 50
Current surrounding social environment:
[{carpenter_0, carpenter_1, carpenter_2, carpenter_3, miner_0, miner_1, miner_2,
    miner_3}].
payoff: 0.1, 0.1, 0.1, 0.1, 0.1, 0.1, 0.2, 0.2
Current surrounding physical environment:
The resources in your observation are: []. The distances of them are [] steps away.
    The numbers of them are [] respectively.

The event grid in your observation are: [Hammer Event]. The distances of them are
    [0] steps away.
The people in your observation are: [miner_0], The distances of them are [1] steps
    away.
Your current inventory:
You have NOTHING in your inventory.

Thoughts: I am carpenter_0, and in the current coalition, there are both carpenters
    and miners. The hammercraft event is right next to me. Since I have no wood and
    no stone, I can also not craft hammer. I don't see any resource in my field of
    view, so I need to explore the map to find one.
Plan: EXPLORE MAP.