# OpenReview forum: "AdaSociety: An Adaptive Environment with Social Structures for Multi-Agent Decision-Making"
_NeurIPS.cc/2024/Datasets_and_Benchmarks_Track — NeurIPS 2024 Track Datasets and Benchmarks Poster_

### Official Review · Reviewer_nGQ4 · 2024-07-17
**Helpful multi-agent simulator; ease of use and claims of mutual adaptation could be improved**

**Rating:** 7
**Confidence:** 4

**Review:**

This paper introduces a highly customizable environment for training and testing multi-agent decision making algorithms under dynamic physical and social dynamics. This is a beneficial environment and testbed to have given the significance of such agents as virtual and physical assistants and agents, particularly due to the advent of LLMs.

Pros

- Highly customizable environment with changing physical dynamics and social structures.
- This environment can hopefully be used by others working on multi-agent decision making algorithms for zero sum games that involve any combination of collaboration, competition, partial observability, centralization vs. decentralization, etc.
- Results showing the shortcomings of state-of-the-art RL algorithms under dynamic social structures, the benefits of curriculum learning, and the potential of LLMs to do well in such settings.
- The paper is well-written and clear in its setup and experimental results.

Cons

- It is unclear how easy or practical it is for users to customize the environment. The authors do not provide a high-level model or API of the environment.
- The claim that the environment also adapts to the agents makes it seem that it involves some element of unsupervised environment design to challenge and improve agents trained in it, which is not the case.
- The Broader Impacts section (and to some extent the Limitations section) should be significantly expanded.

**Strengths:**

- There is a large degree of customization that AdaSociety provides, both in the physical environment (Markov Game components like the state and action spaces, transition dynamics, and reward function; how the Markov Game evolves over time) and in the social structures (agent-agent and agent-organization (hierarchical) connections), as well as how social structures affect reward-sharing and information-sharing. The customizable social structures allow for diverse environments requiring collaboration, competition, contract-formation via negotiation, etc.
- This paper highlights how many RL methods perform poorly out-of-the-box in complex environments, particularly those with complicated and dynamic social structures.

**Additional Feedback:**

- Definition 2
    - Are the universal state and action spaces the largest possible spaces the environment could support? This should be defined more explicitly.
    - Why is it \beta : S_w X A_w and not \beta : S_b X A_b? It seems that \beta should take in the current state and action spaces rather than the universal ones.
    - Why is the output of \beta A_t and not A_{t+1}? The state space is updated but the action space is not?
    - In the output of \beta, why do all the sets have to be strict subsets and not just subsets or not even subsets? It seems that the state and action spaces should be able to remain the same or even contract. Also, they should be able to be as small as the base-MG sets and as large as the universal sets.
- Definition 3
    - Should \beta also be time-dependent? If not, this seems to assume that the MG evolves deterministically and according to a constant function \beta at each time step.
- Lines 282-283: does “Ind.” stand for “independent” and “Ovlp.” for overlapping? If so, this should be stated explicitly and the groups should be explained more clearly. I realize that Figure 2 shows these two groups, but I didn’t see where this was referenced or explained in the text.
- The end of sec. 5.3.1 (lines 293-295) states that the large variance in different policies’ performances suggests that “current algorithms struggle to learn stable policies for dynamic social structure scenarios.” This this variance not just a characteristic of many RL algorithms in general? There are many complex domains without dynamic (or any) social structures where RL also does not learn a stable policy.

**Clarity:**

Yes. Small errors:

- Line 69: “one new resources are synthesized” sounds strange. Perhaps “one new resource is synthesized” or “one new unit of the resource is synthesized” would be better.
- Line 131: “state-of-art” —> “state-of-the-art”
- Line 131: the last sentence starting with “In” is incomplete. It should probably read (line 132) “multiple possible victory paths, requires a balance…” instead of “multiple possible victory paths and requires a balance…”
- Line 151: “various behaviors emergence” —> “the emergence of various behaviors” or “various behaviors emerging”
- Line 164: the sentence starting with “In Growing-MG” contains “in Growing-MG” twice (redundantly), and “some certain transitions” should be reworded.
- Line 171: it would sound clearer to say “$\rho$ is the initial state distribution and $\gamma$ is the temporal discount factor.”
- Definition 1 (line 171): should it be T_b : S_b X A_b X S_b instead of T_b : S X A_b X S_b. Otherwise, it’s not clear what S is. Some arbitrary state space?
- Line 198: “learning coordination with various ones” is unclear. What does “ones” refer to?
- Line 255: “carpenters to execute” —> “carpenters can execute”
- Line 291: “explorations” —> “exploration”
- Line 298: the period after “(See Fig. 7b)” should be a comma. Also, “See” need not be capitalized.
- Line 327: “C/M” should be written out explicitly to avoid confusion.
- Line 356: extra space after “cooperate”
- Line 362: “learn” —> “learns”

**Correctness:**

- The way in which the authors sometimes refer to the environment as AdaSociety adaptively changing the environment misleadingly suggests that it includes some degree of unsupervised environment design, when it does not. Social structures may indeed change depending on how agents form and break connections, but this produces a *dynamic* environment rather than an *adaptive* one. For example, in the abstract, the authors state that “As agents progress, the environment adaptively generates new tasks with social structures for agents to undertake.” While the social structures change, the environment is not adaptively generating new tasks. Similarly, the authors stress the “mutual adaptation between agents and AdaSociety” (line 132). While agents do adapt to the environment, AdaSociety does not actively adapt to the agents’ behaviors; rather, it updates the Markov Game dynamics based on the agents’ actions. This is a nuanced distinction that should be made to avoid claims that AdaSociety does an unsupervised environment design to improve the capabilities of trained agents.
- As mentioned in the Limitations section, it is unclear whether AdaSociety is practically easy to customize and whether it adheres to an API/model of some sort. If not, it would be of limited use to the research community due to the burden of testing out new settings. The paper would benefit from an explanation of how the environment is structures and how one would practically customize it. Appendix A.4 claims that customization is simple, but it is not described in much further detail.

**Documentation:**

Yes, the provided code shows example commands to reproduce the results and examples of how to create new environments.

**Ethics:**

No ethical concerns (as long as the authors adequately expand on the Broader Impact section to describe potential negative impacts).

**Limitations:**

The authors should significantly expand the Broader Impact section. Agents that are trained to navigate and affect social structures, as well as take actions in an environment (whether virtual or physical), may certainly be beneficial, but they also have significant capacity to cause harm. They may learn how to deceive, coerce, and otherwise negatively influence other agents to achieve harmful goals. Even if this is not the intention, reward misspecification and partially observable settings have been shown to produce misaligned and harmful agents, particularly when trained to maximize reward in long-horizon tasks. The authors should discuss these risks in much greater depth.

**Opportunities For Improvement:**

- It is unclear how easy it is to implement new settings/environments and customize the different parameters of AdaSociety. While many aspects of the environment do seem customizable, the authors do not explain how simple this would be to do or whether there is an API. Looking at the provided code, the example environment definitions seem rather extensive. For AdaSociety to be useful to the research community, it should be easy to customize to rapidly test out different physical and social settings.
- An agent’s resource preference, while allowed to be heterogenous, is limited to scale linearly with the amount of the resource in the agent’s inventory. This is a good start, but there are scenarios one can imagine where the reward should not scale linearly. For example, the agent may just care about owning a single item, or the reward may saturate once a particular amount of an item is reached.

**Relation To Prior Work:**

Overall, yes.

It would also make sense to cite Overcooked (Carroll et al, 2019 “On the utility of learning about humans for human-ai coordination”) since it is as environment that involves collaboration and coordination between multiple agents in a physical setting.

**Summary And Contributions:**

- Introduces AdaSociety, a multi-agent environment with support for dynamic physical environments and social structures. This provides a useful simulator for researching multi-agent decision making in a variety of general-sum games, which may be centralized or decentralized and involve combinations of collaboration and competition.
- Provides three settings (”mini-games”) that demonstrate the customization of physical environments and social structures in AdaSociety.
- Tests several popular RL algorithms, and LLM-based approach, and a custom curriculum learning + RL method on the above mini-games.

---

> ### Author Rebuttal · Authors · 2024-08-17
>
> > **Q9:** “Ind.” stand for “independent” and “Ovlp.” for overlapping (Line 282-283)
>
> **A:** Thanks for bringing out the omission of the reference relation. **Yes, here we use "Ind." to represent "Independent" and "Ovlp." to represent "Overlapping".** "Ind. group" only allows each agent to join one group while "Ovlp. group" allows multiple, which could be seen in Figure 2 as well. Due to the limited space in the main manuscript, we provide the details of different types of proposed social structures in the appendix C. **We have added the full words in the revised manuscript.**
>
> > **Q10:** Concerns on the statement "current algorithms struggle to learn stable policies for dynamic social structure scenarios." (Line 293-295)
>
> **A:** Thanks for your comment. We agree that in many complex domains, RL in general cannot learn stable policies, especially MARL. From the experiments in this paper (Figure 3 in Section 5.3 and Figure 9-14 in Appendix E), we found out the social structure setting brings more instability of the RL, compared to "Isolation" and "Connection" scenarios, which do not include groups. Therefore, **we'd like to highlight that the social setting in *AdaSociety* allows researchers not only to investigate scientific questions about social structure/network, complex network, etc. but also to address challenges of using current RL algorithms to obtain meaningful results.** *AdaSociety* provides an environment to develop and improve traditional RL algorithms, specifically for the scenarios where agents have social relationships.
>
> In the manuscript, we revised the statement as "Compared to scenarios without agent groups (Fig. 10 and 11), the results indicate that the current algorithms struggle to learn stable policies for scenarios with agent groups."

---

> ### Author Rebuttal · Authors · 2024-08-17
>
> > **Q4:** Expand the Broader Impact section
>
> **A:** Thanks for your helpful comments. Below is the expanded version of the Broader Impacts section, and we will modify the paper accordingly.
>
> To contribute to the development of multi-agent decision-making algorithms, we propose *AdaSociety*, a customizable environment with massive and diverse tasks generated by expanding state and action spaces and adaptive social structures.  Due to the complexity of tasks and the heterogeneity of agents' capacities and preferences, agents need to team up and even cooperatively establish hierarchical social structures to achieve goals. However, agents may also learn some strategies that are harmful to their co-players, as is common in multi-agent research. We have made significant efforts to mitigate such behaviors through thoughtful design within the environment. Given the heterogeneity among agents and adaptive social structures, harmful behaviors tend to be short-sighted and inferior when it comes to maximizing long-term benefits, with stable cooperation emerging as the optimal strategy. The multiple evaluation metrics introduced in AdaSociety, like fairness, also empower researchers to identify and exclude extreme or exploitative agents and facilitate the learning of cooperative behaviors.
>
> Nevertheless, some harmful behaviors may still arise during training. We ask researchers utilizing our platform to meticulously observe agents' behaviors to ensure they align with human values and preferences. Should any misalignment or misrepresentation happen, we encourage contributions to the source code (including but not limited to new evaluation metrics, environmental dynamics or incentive mechanisms) to enhance the platform.
>
>
>
> > **Q5:** Small Errors
>
> **A:** Thank you very much for your detailed reading. We sincerely thank the reviewer for pointing out these typos and errors. Regarding your questions, it should be $T_b : S_b \times A_b  \times S_b$ in line 171, and "ones" in Line 198 refers to "co-players" or "other agents". We have completed the corresponding revisions in the manuscript.
>
>
>
> > **Q6:** Overcooked should be discussed and cited
>
> **A:** We appreciate your suggestion regarding Overcooked, a well-known multi-agent collaborative environment featuring multiple tasks. In our revised manuscript, we have cited the relevant paper and included a comparison of Overcooked in Table 1.
>
>
>
> > **Q7:** About Definition 2
> >
> > - Are the universal state and action spaces the largest possible spaces the environment could support? This should be defined more explicitly.
> >
> > - Why is it $\beta : S_w \times A_w$ and not $\beta : S_b \times A_b$? It seems that $\beta$ should take in the current state and action spaces rather than the universal ones.
> >
> > - Why is the output of $\beta$ $A_t$ and not $A_{t+1}$? The state space is updated but the action space is not?
> >
> > - In the output of $\beta$, why do all the sets have to be strict subsets and not just subsets or not even subsets? It seems that the state and action spaces should be able to remain the same or even contract. Also, they should be able to be as small as the base-MG sets and as large as the universal sets.
>
> **A:** We have modified Definition 2 according to your comments:
>
> **Definition 2.**  A Monotonic-MG-bundle upon a base-MG $MG_b$ within the universal state and action space $S_w = $ {$S_w^1,\dots,S_w^I$}, $A_w=${$A_w^1,\dots,A_w^I$} is a map $\beta: S_t\times A_t\to ${$S_{t+1}, A_{t+1}, T_{t+1}, R_{t+1}|S_b^i\subseteq S^i_{t}\subseteq S^i_{t+1} \subseteq S^i_w, A^i_b\subseteq A^i_{t}\subseteq A^i_{t+1} \subseteq A^i_w, T_{t+1}\in\mathcal{T}(S_{t+1},A_{t+1}),R_{t+1}\in\mathcal{R}(S_{t+1},A_{t+1})$}.
>
> Here is our response of your questions:
>
> - The universal state space $S_w$ and action spaces $A_w$  represent the largest possible spaces supported by the environment, we will provide the description more explicitly.
>
> - $\beta$ should indeed take the current state $S_t$ and action spaces $A_t$ as inputs. We made the corresponding changes in the revised definition above.
>
> - This involves changes of the transition, where the definition of the transition includes the current state space $S_t$, action space $A_t$, and the next state space $S_{t+1}$. As we noted in the Footnote 2, the initial conditions are $S_0 , A_0, S_1, T_0 \in T(S_0, A_0, S_1)$, so the subscript of the state space is always one step ahead of the action space. Therefore, in each step, $A_t$ and $S_{t+1}$ are updated accordingly.
>
> 	We acknowledge that the time step description here may seem somewhat unintuitive. We believe the revised definition above is more intuitive. To avoid the issue of needing two time steps to define the transition function, we can modify the transition from $\mathcal{T}(S,A,S')=${$T|T:S\times A\to S'$} to $\mathcal{T}(S,A)=${$T|T:S\times A\to S$}, meaning that the transition function is required to be closed within the state space S. In this case, we can define the bundle as $\beta: S_t \times A_t \to ${$S_{t+1}, A_{t+1}, T_{t+1}, R_{t+1}$} (omitting the subset constraint). For a detailed definition, please refer to the revised paper.
>
> - We greatly appreciate the reviewer pointing out this issue. Our paper emphasizes the dynamic expansion of state and action spaces, where the mathematical expression should represent a subset, not necessarily a strict subset. We have made the necessary corrections in the revised definition.
>
> > **Q8:** Should $\beta$ be time-dependent? (Definition 3)
>
> **A:** As we mentioned in the above explanation and in Footnote 1 of the paper, $\beta$ is indeed also a function of time. In the alternative formulation we provided above, $\beta$ takes the current time step's state and action as inputs, so it is time-dependent.

---

> ### Author Rebuttal · Authors · 2024-08-17
>
> > **Q1:** More detailed explanation and instructions for customization
>
> > **Q2:** Customization of any reward function rather than linear reward function
>
> **A:** Thanks for your insightful comment! We appreciate the emphasis on customization within *AdaSociety*. We fully agree with you that supporting various customization, including more diverse reward structures, will enhance the environment's flexibility and utility in fostering various social intelligence.
>
> **We have provided more detailed instructions for each mini-game in our repository, located in `/docs`.** In general, Our platform offers interfaces with varying degrees of flexibility to accommodate different customization levels. Training parameters and some environmental parameters (such as mini-game selection, map size, episode length, and negotiation phase length) can be configured through the shell. More complex environmental parameters (like the initial position generation method for each entity, inventory capacity and value preference of each agent, and non-built-in event configurations) require input through a Python file located in `project/args`. The required content and specific format for this file are outlined in the documentation, along with examples in `project/args`. For further development, such as defining different semantics for edges in social components or creating unique stages, you can modify the underlying code in `project/Environment/Map.py` and follow the examples in `project/Environment/example`.
>
> **We are also developing a new version to enhance customizability and modularity.** This version aims to use configuration files in JSON to manage most configurations, including training parameters, environment settings, and new mini-game implementations. Additionally, we plan to provide Agent modules with APIs for convenient customization, including the diverse reward structures you mentioned. (Considering the variety of possible reward functions, these are better implemented through APIs rather than configuration files. However, configuration files will still support linear preference settings.) We are trying our best to release this new version before the author-reviewer discussion phase ends.
>
> Thank you again for your valuable feedback. Your insights are instrumental in guiding our improvements to *AdaSociety*.
>
>
>
> > **Q3:** Adaptation of the environment
>
> **A:** Thanks for your valuable comments!
>
> In the paradigm of Unsupervised Environment Design (UED) [1,2,3], the environment learns a policy $\Gamma: \Pi \to \Delta(\Theta^T)$, which is a function from agent policy $\Pi$ to the environment’s parameters $\Theta^T$. Such a policy will automatically produce a distribution over solvable environments and further support the continued learning of agent’s policy. *AdaSociety* does not implement an unsupervised environment design to produce diverse tasks. *AdaSociety* has no goals or objectives, like most ecological systems, and produces multiple tasks through adaptive social structures and expanding physical surroundings. **We will add UED in the Section of Related Works and discuss the difference between UED and *AdaSociety*.**
>
> Then, we would like to explain why *AdaSociety* is referred to as an adaptive environment.
> In complex network theory, a network is called an adaptive network, if there is a feedback loop between the attributes or behavior of nodes and the topology of the network [4,5,6]. In *AdaSociety*, agents build or break connections with others and impact social structure. Conversely, social structure influences agents’ observations and reward structures and further influences their attributes and behavior. Thus, following the definition of adaptive networks, the social structure of *AdaSociety* is adaptive. As a key component of *AdaSociety*, social structure influences the generation of new tasks. For example, independent agents collect all kinds of available resources to synthesize high-level resources. However, the team-up agents will be mostly rewarded by collecting or synthesizing some specific kind of resources, according to the division of labor in the team. Furthermore, agents initially can only observe very limited resources (wood and stone in our mini-games) and events (Hammercraft). Through exploration in *AdaSociety*, agents gradually discover new resources and events. The appearance of a kind of new resources depends on agents’ behavior. For instance, as shown by the synthesis tree in Figure 4, which appears next, shovel or cutter, depends on agents’ behavior. To sum up, we say *AdaSociety* is an adaptive environment.
>
> We hope our explanation can alleviate your concerns. If you have any further questions, please feel free to ask! We are happy to discuss with you!
>
> [1] Dennis, M., et al. (2020). Emergent complexity and zero-shot transfer via unsupervised environment design. Advances in neural information processing systems, 33, 13049-13061.
>
> [2] Mediratta, I., et al. (2023, November). Stabilizing unsupervised environment design with a learned adversary. In Conference on Lifelong Learning Agents (pp. 270-291). PMLR.
>
> [3] Jiang, M., et al. (2022). Grounding aleatoric uncertainty for unsupervised environment design. Advances in Neural Information Processing Systems, 35, 32868-32881.
>
> [4] Gross, T., & Blasius, B. (2008). Adaptive coevolutionary networks: a review. Journal of the Royal Society Interface, 5(20), 259-271.
>
> [5] Moreno-Mateos, D.,et al. (2020). The long-term restoration of ecosystem complexity. Nature Ecology & Evolution, 4(5), 676-685.
>
> [6] Berner, R., et al. (2021). Desynchronization transitions in adaptive networks.Physical Review Letters, 126(2), 028301.

---

### Official Review · Reviewer_vq28 · 2024-07-24
**Review comments**

**Rating:** 6
**Confidence:** 3
**Correctness:** Yes
**Clarity:** Yes

**Review:**

The paper is generally well-written and easy to follow. The introduction of an alterable social structure, which influences the rewarding function, is a novel contribution to RL environments. However, it would be beneficial to provide detailed explanations of the social structure, including the distinction between community-level and group-level, as well as the rationale behind selecting three levels. Additionally, the inclusion of more complexity levels in the experiments would enhance the demonstration of the scalability of challenges posed by the social structure. Furthermore, improving the user-friendliness of the submitted code by providing detailed installation instructions, API documentation, and proofreading the accompanying document would be valuable for reproducibility.

Overall, the paper presents valuable insights into a promising research direction, shedding light on the further development of multi-agent methods in understanding physical dynamics and social cooperation. However, additional details and improvements would enhance the quality and comprehensibility of the paper.

**Strengths:**

1. The paper is easy to follow and well written.

2. The introduction of alterable social structure which influent the rewarding function is novel in RL environments.

**Additional Feedback:**

N/A

**Documentation:**

Could be improved.

**Limitations:**

See improvements.

**Opportunities For Improvement:**

1. Some detailed explanations of the social structure can be included. For example, the authors can clarify the difference between community-level and group-level structures. Additionally, they can provide the rationale behind choosing three levels for the social structure.

2. The current design of only two complexity levels for each mini-game may not sufficiently demonstrate the scalability of challenges provided by the social structure. It would be more persuasive if the experiments included more levels to showcase the system's ability to handle increasingly complex scenarios.

3. To make the submitted code more user-friendly, the authors should provide a detailed description of the installation process, API documentation, and other relevant instructions. It is also advisable for the authors to proofread the document again to avoid any unnecessary inconvenience in reproducing their work. For instance, they mentioned "check an example in project/advanced_args.py" in README.md, but the actual file may be located in the project/args folder.

**Minor Questions**
1. In Line 190, the authors wrote "environmental and social... to include observational information." Is this one of the configurations in AdaSociety? How is it implemented in the environment?
2. In Line 232, what is the reason for setting $w_i + w_j \leq 1$ instead of $w_i + w_j = 1"? Alternatively, why not allow the agent to choose $w_i \leq 1$ and calculate $w_j = 1 - w_i$ directly?
4. In Line 303, what evidence supports the authors' claim that agents "may forget the strategies..."?
5. In Line 346, could you provide a more detailed explanation of the term "oracle policy"?

**Relation To Prior Work:**

Yes

**Summary And Contributions:**

This paper introduces a new environment for evaluating the performance of multi-agent methods in understanding physical dynamics and social cooperation.The proposed AdaSociety is characterized by an explicit and alterable social structure represented as multi-layer directed graph, and allows agents to alternate the social structure at any time in the game play via "social actions". The authors conducted experiments on three predefined mini-games to compare LLM-based and some state-of-the-art multi-agent methods. The experimental results indicate that the methods above still have spaces for performance improvement, which points out a possible direction for follow-up research in the multi-agent field.

---

> ### Author Rebuttal · Authors · 2024-08-17
>
> > **Q5:** $w_i+w_j=1$ vs $w_i+w_j\leq1$ (Line 232)
>
> **A5:** This is a typo. The correct setting should be $w_i+w_j=1$. We apologize for the confusion. Thanks for pointing it out.
>
> > **Q6:** The evidence of CL agents "may forget the strategies" in *Contract* (Line 303)
>
> **A6:** The rationale behind our statement lies in the group reward during different phases of curriculum learning. At the end of Curriculum 1, the group's reward peaked, indicating successful learning within the physical environment at that stage. The group's reward exhibited a gradual decline during Curriculum 2. While in Contract-easy, agents managed to re-enhance the reward in Curriculum 3, such recovery was not observed in Contract-Hard. This led us to speculate that the agents might have forgotten the strong policy acquired during Curriculum 1 when facing the challenges of Contract-Hard. The following is a table of group rewards after each curriculum:
>
> ||*Contract-Easy*|*Contract-Hard*|
> |-|-|-|
> |Cur. 1|0.9747$\pm$0.0059|0.6470$\pm$0.0313|
> |Cur. 2|0.3435$\pm$0.0262|0.2566$\pm$0.0284|
> |Cur. 3|0.9136$\pm$0.0023|0.2773$\pm$0.0466|
>
> We acknowledge the need for empirical evidence to substantiate this claim. We will add the above table for the entire CL training process to the revised manuscript to support our claim. Thank you for your question!
>
> > **Q7:** A more detailed explanation of the term "oracle policy" (Line 346)
>
> **A7:** Thanks for pointing out the unclear definition. The "oracle policy" refers to the optimal solution that maximizes the total credits obtained by agents with limited resources. The optimal solution describes the final inventories of each type of resources, as well as the number of performed synthesizing events. Therefore, we regard the credits of oracle policy as the maximal rewards in the environment, and use it as a normalizer to evaluate the performance of the decision-making methods. In Supplementary Material, we provide a Mixed Integer Linear Programming (MILP) model to calculate the oracle policy. Since this MILP model calculates the theoretically maximal rewards, without losing precision, we assume agents can perform all the events and collect all the resources, and neglects the timesteps spent on searching for or approaching resources and event grids.

---

> ### Author Rebuttal · Authors · 2024-08-17
>
> > **Q1:** Detailed explanations of the social structure
>
> **A1:** We are sorry for the unclarity of the description of social structure. Figure 6, which shows a three-level social structure, is a schematic diagram. **Actually, *AdaSociety* supports the description of social structures with arbitrary levels, depending on the research problems and the required granularity of social structures.**
>
> We recognize that the definitions of “group-level” and “community-level” are inaccurate and may cause misconception. Thus, they have been modified as “1st-level” and “2nd-level” of the social structure, respectively. And the “individual-level” is the “0th-level”. It consists of individual agents, who are the fundamental units of decision-making. Any agent (represented as a node) on the kth-level (k>=1) is composed of its connected agents on the (k-1)th-level. Its decision-making relies on group norms, like voting, consensus decision-making and delegation. A kth-level agent will affect its (k-1)th-level neighbors’ reward functions and observations, thereby influencing their decision-making and enabling their division of labour and cooperation. One agent on the (k-1)th-level may be simultaneously subordinate to any number of agents on the kth-level. For example, an individual employee is the 0th-level agent, a project team composed of several employees is the 1st-level agent, a company consisting of many teams is the 2nd-level agent, and a business group composed of many companies is the 3rd-level agent.
>
> Thanks again for your valuable comments! We will modify the manuscript based on your suggestions.
>
> > **Q2:** More complexity levels
>
> **A2:** Besides the three mini-games with two complexity levels: Social structure, Contract and Negotiation, we have provided another mini-game, Exploration, showing greater complexity. Exploration is mentioned is Section 4, and a detailed description is given by Subsection C.2 and C.3 in the Appendix. In Exploration, all the built-in resources and events are included. Physical actions and social actions are available at every timestep. In particular, Exploration provides at most nine tasks:  HammerCraft, TorchCraft, SteelMaking, Potting, ShovelCraft, PickaxeCraft, CutterCraft, GemCutting and TotemMaking (see details in Table 5). In Exploration, agents need to elaborate more exploration, decision-making and planning abilities to achieve better outcomes.
>
> We also conducted experiments to evaluate the speed across various complexities in three mini-games. Specifically, we tested five different levels of complexity, employing 4 (Easy task), 8 (Hard task), 20, 100, and 1000 agents, and the number of group nodes is the same as the number of agents. All agents operated under a random policy, and the tests were executed in a single process with only one environment active.
>
> The following table shows the average number of game steps that can be taken per second (all players will perform an action in each game step):
>
> ||4p|8p|20p|100p|1000p|
> |-|-|-|-|-|-|
> |*Negotiation*|1273.40|532.58|314.18|88.32|7.04|
> |*Contract*|1264.56|533.50|347.61|104.39|11.72|
> |*Social Structure - Dynamic*|1213.06|507.50|294.30|63.25|4.90|
>
> **Our results indicate that the speed of the environment is approximately inversely proportional to the number of agents and groups.** In other words, the time required for each step tends to increase linearly with the number of agents and groups. **We think the results show that the environment itself operates efficiently within a reasonable timeframe.** The inference of the neural network is the bottleneck when implementing neural network-based policies.  Additionally, with a larger number of agents, the speed may be constrained by parallel processing strategies and available computing resources.
>
> This indicates that *AdaSociety* could handle incresasingly complex scenarios. However, traditional RL methods, including PPO, RecPPO, MAPPO and Rainbow, do not perform well in this highly complex mini-task, as offered in Table 8. We speculate that agents only explore some basic tasks/events, like HammerCraft and TorchCraft and do not learn to build social relationships to obtain higher rewards.
>
> > **Q3:** A detailed description of the documentation and instructions
>
> **A3:** We appreciate your detailed reading of the document and pointing out some errors. We apologize for the typos in the documents. **We have reviewed the documents and provided more detailed instructions for each mini-game in our repository, located in `/docs`. General instructions and the instructions specific to LLM are also provided in `/docs`.**
>
> We are also developing a new version to enhance customizability and modularity. This version aims to use configuration files in JSON to manage most configurations, including training parameters, environment settings, and new mini-game implementations. Additionally, we plan to provide Agent modules with APIs for convenient customization. We are trying our best to release this new version before the author-reviewer discussion phase ends.
>
> > **Q4:** The implementation of observational information (Line 190)
>
> **A4:** In our environment, we generate partial observations by masking the state. Specifically, the `obs_range` parameter in `rllib_train.py` defines the size of the observation domain; any information outside this range is not passed to the agent. Additionally, if the agent does not meet specific resource or event requirements (detailed in Appendix A.1 and A.2), those resources or events will also be masked from the agent's observation, even if they are present on the map. Partial observation of the social state is implemented through similar masking techniques. However, current experiments provide all agents with complete social state information. We believe that partially observable social states is more appropriate for tasks that involve greater complexity and a larger number of agents.

---

### Official Review · Reviewer_CKwz · 2024-07-25
**A comprehensive and customizable multi-agent environment**

**Rating:** 7
**Confidence:** 4
**Correctness:** Yes
**Clarity:** Yes

**Review:**

Pros:
1. AdaSociety's integration of adaptive physical and social components presents a novel platform for multi-agent research.
2. The environment generates a wide variety of tasks dynamically.
3. Researchers can tailor the environment to specific needs, allowing for extensive experimentation and development of new algorithms.
4. The inclusion of well-defined mini-games provides clear benchmarks for evaluating the performance of different algorithms.
5. The paper provides a comprehensive set of baseline results using both RL and LLM-based methods.
6. The manuscript is clearly written and easy to follow.

Cons:
1. The environment's complexity might pose scalability challenges, particularly when extending to larger agent populations or more intricate social structures.
2. The time complexity is not evaluated.
3. While diverse metrics are provided, the paper could benefit from a more detailed analysis of how these metrics correlate with real-world applications or theoretical insights.

**Strengths:**

AdaSociety's integration of adaptive physical and social components presents a novel platform for multi-agent research, generating a wide variety of tasks dynamically. Researchers can tailor the environment to specific needs, allowing for extensive experimentation and the development of new algorithms. The inclusion of well-defined mini-games provides clear benchmarks for evaluating the performance of different algorithms. Additionally, the paper offers a comprehensive set of baseline results using both RL and LLM-based methods, and the manuscript is clearly written and easy to follow.

**Additional Feedback:**

N/A

**Documentation:**

Yes

**Limitations:**

Yes

**Opportunities For Improvement:**

The environment's complexity might pose scalability challenges, particularly when extending to larger agent populations or more intricate social structures. Additionally, the time complexity is not evaluated, and while diverse metrics are provided, the paper could benefit from a more detailed analysis of how these metrics correlate with real-world applications or theoretical insights.

**Relation To Prior Work:**

Yes

**Summary And Contributions:**

This manuscript introduces AdaSociety, a multi-agent environment designed to enhance decision-making capabilities by integrating adaptive physical surroundings and dynamic social structures. AdaSociety continuously generates new tasks influenced by agents' interactions and social connections. The environment features three mini-games that highlight various social structures, demonstrating that specific social configurations can benefit individual and collective agent performance. Preliminary results using reinforcement learning and LLM algorithms reveal the potential and current limitations of leveraging social structures.

---

> ### Author Rebuttal · Authors · 2024-08-17
>
> > **Q3:** Analysis of how the metrics correlate with real-world applications or theoretical insights.
>
> **A3:** Thanks for your comments!
>
> As an open platform for multi-agent decision-making, *AdaSociety* provides multiple evaluation metrics to guide the learning of algorithms and offer insights into agents' behaviors.
>
> **Individual reward** is one of the most common metrics for decision-making problems. It measures agents' decision-making abilities in maximizing self-interest. However, relying solely on individual rewards can be risky. In general-sum games, agents focus on maximizing their own rewards may engage in shortsighted and exploitative behaviors that harm their own long-term rewards and the collective benefit. For example, in Prisoner's Dilemma, self-interested agents always fall into the inefficient Nash equilibrium of defection, which minimizes one's own reward and the collective benefit. To tackle this issue, we introduce the **fairness** score calculated using the Gini index, which evaluates fairness within a group. In real societies, fairness is a crucial component of social justice, significantly influencing the stability of social structures and the maintenance of long-term cooperation. This metric serves as a reference for selecting agents and algorithms that balance efficiency and fairness, rather than merely pursuing individual gains.
>
> The **completion rate** is introduced to measure agents' exploration within the synthesis tree. It is calculated as the ratio of actual executions of events to the optimal executions by the oracle policy (computation of the oracle policy can be found in Supplementary).  A higher number of events with non-zero completion rates indicates deeper exploration, while a completion rate closer to 1 reflects a more effective policy.  Exploration is crucial in RL. The introduction of **completion rate** will guide decision-making algorithms avoid local optima, actively explore the environment, and find the optimal policy effectively.
>
> Social structure is the distinctive feature of *AdaSociety*. Degree-based metrics, including **average degree** and **maximum degree**, are proposed to describe and measure the topology of social structure, which significantly influences agents' policies and performances by shaping their information streams and reward functions. Agents' degree distribution is generally correlated with their rewards. For example, an agent with high degree can obtain more information or participate in more reward distribution, thereby gaining higher returns. Combining degree-based metrics with other metrics, like individual reward and fairness, we can recognize the effective social structure for scenarios, guiding the learning of algorithms.
>
> Thanks again for your helpful comments! We will further clarify our definition of evaluation metrics and add the above analysis into our revised manuscript.

---

> ### Author Rebuttal · Authors · 2024-08-17
>
> > **Q1:** Scalability Challenges
>
> **A1:** Thank you for your feedback regarding the scalability of our environment. We conducted experiments to evaluate the speed across various complexities in three mini-games. Specifically, we tested five different levels of complexity, employing 4 (Easy task), 8 (Hard task), 20, 100, and 1000 agents, and the number of group nodes is the same as the number of agents. All agents operated under a random policy, and the tests were executed in a single process with only one environment active.
>
> The following table shows the average number of game steps that can be taken per second (all players will perform an action in each game step):
>
> ||4p|8p|20p|100p|1000p|
> |-|-|-|-|-|-|
> |*Negotiation*|1273.40|532.58|314.18|88.32|7.04|
> |*Contract*|1264.56|533.50|347.61|104.39|11.72|
> |*Social Structure - Dynamic*|1213.06|507.50|294.30|63.25|4.90|
>
> **Our results indicate that the speed of the environment is approximately inversely proportional to the number of agents and groups.** In other words, the time required for each step tends to increase linearly with the number of agents and groups. **We think the results show that the environment itself operates efficiently within a reasonable timeframe.** The inference of the neural network is the bottleneck when implementing neural network-based policies.  Additionally, with a larger number of agents, the speed may be constrained by parallel processing strategies and available computing resources.
>
>
>
> > **Q2:** Evaluating the time complexity
>
> **A2:** We fully agree that a comprehensive evaluation needs to be combined with time complexity. In the table below, we present the time and number of game steps taken by various learning algorithms to achieve convergence across different scenarios. It is important to note that the specific time metrics are influenced by resource allocation, algorithm implementation, and parallelism strategy. Consequently, the relative ratios hold more significance than the absolute values. Given that resource usage may vary across tasks, the time comparisons in each column may not provide meaningful insights. Additionally, evaluating convergence time should be coupled with the rewards achieved for a comprehensive assessment of each algorithm's performance.
>
> The following table outlines the time (hours) and number of game steps taken by the algorithms to converge:
>
> |Time (hours) \ Game Steps|CL|PPO|RecPPO|MAPPO|Rainbow|
> |-|-|-|-|-|-|
> |Negotiation-Easy|5.66$\pm$0.16 \ 14M|0.12$\pm$0.03 \ 0.47M|1.05$\pm$0.04 \ 2.4M|21.21$\pm$0.35 \ 1.8M|14.97$\pm$0.30 \ 42M|
> |Negotiation-Hard|12.96$\pm$0.39 \ 15M|1.66$\pm$0.03 \ 2.3M|0.30$\pm$0.01 \ 0.49M|74.85$\pm$9.75 \ 4.42M|40.54$\pm$0.54 \ 53M|
> |Contract-Easy|32.03$\pm$1.38 \ 112M|0.19$\pm$0.03 \ 0.80M|1.94$\pm$0.03 \ 6.0M|9.65$\pm$0.02 \ 1.2M|9.65$\pm$1.49 \ 25M|
> |Contract-Hard|48.58$\pm$0.56 \ 78M|0.21$\pm$0.02 \ 0.56M|0.74$\pm$0.01 \ 1.0M|14.94$\pm$0.01 \ 0.94M|19.07$\pm$1.25 \ 37M|
> |Social Structure - Dynamic|6.98$\pm$0.64 \ 4.8M|6.90$\pm$0.55 \ 6.0M|10.16$\pm$0.12 \ 6.0M|46.23$\pm$1.21 \ 4.8M|2.34$\pm$33.76 \ 2.0M|
>
> Overall, PPO exhibits the fastest convergence speed and average runtime, closely followed by RecPPO. While MAPPO's number of game steps is comparable to that of PPO, its average time per step is significantly longer due to its centralized training approach, resulting in extended total convergence time. Conversely, CL requires a longer training duration and more steps to achieve rewards that surpass those of other algorithms. Rainbow, while taking many steps to converge, ultimately yields subpar rewards. Although Rainbow shows fewer steps in the Social Structure scenario, it achieves a considerably low reward (see Fig. 3(c)). We will add the data and analysis to our revised manuscript.

---

### Author Rebuttal · Authors · 2024-08-31

We would like to thank the reviewers for recognizing our environment as novel (CKwz, vq28, nGQ4), highly customizable (CKwz, vq28, nGQ4), and insightful for future research (vq28, nGQ4), and equipped with clear benchmarks for evaluation and comprehensive baseline results (CKwz). We are also pleased to hear that the reviewers found our manuscript to be clearly written and easy to follow (CKwz, vq28, nGQ4).

We have addressed each reviewer’s questions individually and hope our responses adequately resolve your concerns.

We are excited to announce that we have completed the latest version of *AdaSociety* and updated our source code, available at the link provided in the abstract of the article. This version focuses on making the customization process clearer and simpler, allowing users to navigate our environment more easily.

Users can now utilize JSON files to configure all environment parameters, including the initial map generation, player heterogeneity, social structure scheduling, and post-update logic. Additionally, we have distinctly separated the environment framework from the training framework. Users can create their own *Environment Handler* and *Agent* in Python to connect *AdaSociety* with their preferred training platform or algorithm. We have also provided code to connect *AdaSociety* with the well-known RL library *RLlib*.

Users can implement their own observation functions, reward functions, and action construction within the *Agent* class. This, combined with JSON configuration, allows users to design their own mini-games. To further assist users, we have provided detailed documentation in the repository.

We invite the reviewers to explore the latest version of our code. If you have any further questions or concerns, please feel free to reach out. Thank you for your time and effort!

---

### Decision · Program_Chairs · 2024-09-26

**Decision:**

Accept (Poster)

**Comment:**

AdaSociety is a multi-agent environment designed to enhance decision-making by integrating adaptive physical surroundings and dynamic social structures.  It can be used for diverse scenarios involving collaboration, competition, and decentralized systems. Experiments using reinforcement learning and LLM-based methods show the potential of leveraging these social structures for improved agent performance, and the paper points out a possible direction. The paper is well-written and presents novel insights.